# Orthogonal Representation Learning for Estimating Causal Quantities

## Abstract

Representation learning is widely used for estimating causal quantities (e.g., the conditional average treatment effect) from observational data. While existing representation learning methods have the benefit of allowing for end-to-end learning, they do not have favorable theoretical properties of Neyman-orthogonal learners, such as double robustness and quasi-oracle efficiency. Also, such representation learning methods often employ additional constraints, like balancing, which may even lead to inconsistent estimation. In this paper, we propose a novel class of Neyman-orthogonal learners for causal quantities defined at the representation level, which we call OR-learners. Our OR-learners have several practical advantages: they allow for consistent estimation of causal quantities based on any learned representation, while offering favorable theoretical properties including double robustness and quasi-oracle efficiency. In numerous experiments, we show that, under certain regularity conditions, our OR-learners improve existing representation learning methods and achieve state-of-the-art performance. To the best of our knowledge, our OR-learners are the first work to provide a unified framework of representation learning methods and Neyman-orthogonal learners for causal quantities estimation.

## 1 Introduction

Estimating causal quantities has many applications in medicine (Feuerriegel et al., 2024), policy-making (Kuzmanovic et al., 2024), marketing (Varian, 2016), and economics (Basu et al., 2011). Here, different causal quantities are of interest such as the conditional average treatment effect (CATE) and the conditional average potential outcomes (CAPOs). For example, in personalized medicine, CATE estimation can help in predicting the relative benefits of different treatment options, so that the one with the best health outcome is selected.

Recently, representation learning methods have gained wide popularity in estimating causal quantities from observational data (e.g., Johansson et al., 2016; Shalit et al., 2017; Hassanpour & Greiner, 2019a;b; Zhang et al., 2020; Assaad et al., 2021; Johansson et al., 2022). One benefit of representation learning methods is that they allow for *end-to-end* learning. Specifically, these methods aim to learn low-dimensional representations where sometimes additional constraints are enforced to tackle inherently causal inductive biases. This typically helps to reduce the estimation variance, especially in low-sample low-overlap settings. For example, *balancing* is a common constraint to reduce the influence of instrumental variables among the covariates (Johansson et al., 2022), which helps to improve the finite-sample performance when the data-generating mechanism indeed has many instruments. Similarly, disentanglement aims to address an inductive bias that different nuisance functions might share or not share common information.

However, constraints on representations can be problematic: constrained representations can lose their asymptotic validity when too strict constraints are applied and estimation becomes inconsistent. This phenomenon is also known as *representation-induced confounding bias* (Johansson et al., 2019; Melnychuk et al., 2024). As a remedy, we later present a framework to quasi-oracle efficiently (and, thus, consistently) estimate causal quantities even based on asymptotically invalid representations.

A related literature stream seeks to estimate causal quantities through a model-agnostic framework of Neyman-orthogonal learners. Prominent examples are the DR-learners and the R-learner (Vansteelandt & Morzywołek, 2023; Morzywołek et al., 2023). They usually split estimation into two stages:

Figure 1: Overview of the connections between representation learning and the estimation of causal quantities. (i) Representation learning can help in estimating causal quantities by providing tools to address different causal inductive biases (e. g., balancing, invertibility, and disentanglement). Conversely, (ii) the estimation of causal quantities can be performed based on general-purpose constrained representations (e. g., fair representations or representations that are learned in an un-/self-supervised way). Our *OR-learners* can be used in both cases.

nuisance functions estimation and target model fitting, and, notably, any machine learning model can be employed at each of the stages. Unlike end-to-end representation learning, Neyman-orthogonal learners offer several favorable theoretical properties. For example, under regularity conditions, Neyman-orthogonal learners guarantee double robustness and quasi-oracle efficiency in asymptotic regime (Chernozhukov et al., 2017; Foster & Syrgkanis, 2023). Further, by employing a separate target model in the second stage, Neyman-orthogonal learners help to address another causal inductive bias, namely that the ground-truth CATE function can be "simpler" than individual CAPOs (Curth & van der Schaar, 2021a). Yet, the connections between Neyman-orthogonal learners and the end-to-end representation learning methods are still not well understood.

In this paper, we unify two streams of work, namely, representation learning methods and Neyman-orthogonal learners. Specifically, we propose a novel, general framework to perform an asymptotically quasi-oracle efficient (and, thus, consistent) estimation of causal quantities based on the learned representations, which we call *orthogonal representation learners* (*OR-learners*). Our *OR-learners* are highly flexible as they target at estimating different causal quantities, like CAPOs and CATE, at the representation level of heterogeneity (Fig. 1). Furthermore, our *OR-learners* effectively solve the drawbacks of constrained representations (i.e., representation-induced confounding bias caused by too strict constraints) and bring favorable theoretical properties associated with Neyman-orthogonality, namely, double robustness and quasi-oracle efficiency.

In sum, **our contributions** are as follows:[1] **(1)** We introduce the *OR-learners*, a novel framework to unify representation learning methods and Neyman-orthogonal learners. **(2)** We show theoretically that our *OR-learners* address the drawbacks of existing end-to-end representation learning methods. That is, our *OR-learners* allow us to perform a quasi-oracle efficient estimation of causal quantities while offering other favorable properties related to Neyman-orthogonality. **(3)** We demonstrate that, under regularity conditions, our *OR-learners* improve the performance in estimating causal quantities for existing representation learning methods.

## 2 RELATED WORK

Our work aims to unify two streams of work, namely, representation learning methods and Neyman-orthogonal learners. We briefly review both in the following (see the full overview in Appendix A).

**Representation learning for estimating causal quantities.** Several methods have been previously introduced for *end-to-end* representation learning of CAPOs/CATE (see, in particular, the seminal works by Johansson et al., 2016; Shalit et al., 2017; Johansson et al., 2022). A large number of works later suggested different extensions to these. Existing methods fall into three main streams: (1) One can fit an *unconstrained shared representation* to directly estimate both potential outcomes surfaces (e.g., **TARNet** Shalit et al., 2017). (2) Some methods additionally enforce a *balancing constraint based on empirical probability metrics*, so that the distributions of the treated and untreated representations become similar (e.g., **CFR** and **BNN** Johansson et al., 2016; Shalit et al., 2017). Importantly, balancing based on empirical probability metrics is only guaranteed to perform a consistent estimation for *invertible* representations since, otherwise, balancing leads to a *representation-induced confounding bias* (RICB) (Johansson et al., 2019; Melnychuk et al., 2024). Finally, (3) one can additionally perform *balancing by re-weighting* the loss and the distributions of the representations with learnable weights (e.g., **RCFR** Johansson et al., 2022). We later adopt the representation learning methods from (1)–(3) as baselines.

---

[1]Code is available at `https://anonymous.4open.science/r/OR-learners`.

**Neyman-orthogonal learners**. Causal quantities can be estimated using model-agnostic methods, so-called *meta-learners* (Künzel et al., 2019). Prominent examples are the R-learner (Nie & Wager, 2021) and DR-learner (Kennedy, 2023; Curth et al., 2020). Meta-learners have several practical advantages (Morzywolek et al., 2023): (i) they oftentimes offer favorable theoretical guarantees such as Neyman-orthogonality (Chernozhukov et al., 2017; Foster & Syrgkanis, 2023); (ii) they can address the causal inductive bias that the CATE is "simpler" than CAPOs (Curth & van der Schaar, 2021a), and (iii) the target model obtains a clear interpretation as a projection of the ground-truth CAPOs/CATE on the target model class. Curth & van der Schaar (2021b) provided a comparison of meta-learners implemented via neural networks with different representations, yet with the target model based on the original covariates (the representations were only used as an interim tool to estimate nuisance functions). However, in our work, we study the learned representations as primary inputs to the target model.

**Research gap.** Our work is the first to unify representation learning methods and Neyman-orthogonal learners. As a result, one can combine any representation learning method from above with our *OR-learners*, which then (i) offer favorable properties of Neyman-orthogonality and (ii) address the causal inductive bias that the CATE is "simpler" than CAPOs.

## 3 PRELIMINARIES

**Notation.** We denote random variables with capital letters $Z$, their realizations with small letters $z$, and their domains with calligraphic letters $\mathcal{Z}$. Let $\mathbb{P}(Z)$, $\mathbb{P}(Z = z)$, $\mathbb{E}(Z)$ be the distribution, probability mass function/density, and expectation of $Z$, respectively. Let $\mathbb{P}_n\{f(Z)\} = \frac{1}{n}\sum_{i=1}^n f(z_i)$ be the sample average of $f(Z)$, and $\|\cdot\|_{L_2}$ be the $L_2$-norm with $\|f\|_{L_2} = \sqrt{\mathbb{E}(f(Z)^2)}$. Then, we define the following nuisance functions: $\pi_a^x(x) = \mathbb{P}(A = a \mid X = x)$ is the *covariate propensity score* for the treatment $A$, and $\mu_a^x(x) = \mathbb{E}(Y = y \mid X = x, A = a)$ is the *expected covariate-conditional outcome* for the outcome $Y$. Similarly, we define $\pi_a^\phi(x) = \mathbb{P}(A = a \mid \Phi(X) = \phi)$ and $\mu_a^\phi(\phi) = \mathbb{E}(Y = y \mid \Phi(X) = \phi, A = a)$ as the *representation propensity score* and the *expected representation-conditional outcome* for a representation $\Phi(x) = \phi$, respectively. Importantly, the upper indices in $\pi_a^x, \mu_a^x, \pi_a^\phi, \mu_a^\phi$ indicate whether the corresponding nuisance functions depend on the covariates $x$ or on the representation $\phi$. In, our work, we adopt the standard Neyman-Rubin potential outcomes framework (Rubin, 1974). Let $Y[a]$ be the *potential outcome* after intervening on the treatment $do(A = a)$, and let $Y[1] - Y[0]$ be the *treatment effect*.

**Problem setup.** To estimate the causal quantities, we make use of an observational dataset $\mathcal{D}$ that contains high-dimensional covariates $X \in \mathcal{X} \subseteq \mathbb{R}^{d_x}$, a binary treatment $A \in \{0, 1\}$, and a continuous outcome $Y \in \mathcal{Y} \subseteq \mathbb{R}$. For example, a common setting is anti-cancer therapy, where the outcome is the tumor growth, the treatment is whether chemotherapy is administered, and covariates are patient information such as age and sex. The dataset $\mathcal{D} = \{x_i, a_i, y_i\}_{i=1}^n$ is assumed to be sampled i.i.d. from a joint distribution $\mathbb{P}(X, A, Y)$, where $n$ is the dataset size.

**Causal quantities.** We are interested in the estimation of two major causal quantities at the covariate level of heterogeneity: *conditional average potential outcomes (CAPOs)* given by $\xi_a^x(x)$, and the *conditional average treatment effect (CATE)* given by $\tau^x(x)$, with

$$\xi_a^x(x) = \mathbb{E}(Y[a] \mid X = x) \quad \text{and} \quad \tau^x(x) = \mathbb{E}(Y[1] - Y[0] \mid X = x) = \xi_1^x(x) - \xi_0^x(x). \tag{1}$$

The estimation of causal quantities can be alternatively formulated as a mean squared error (MSE) minimization task:

$$\xi_a^x(\cdot) = \underset{g \in \mathcal{G}}{\arg\min}\, \mathbb{E}(Y[a] - g(X))^2 \quad \text{and} \quad \tau^x(\cdot) = \underset{g \in \mathcal{G}}{\arg\min}\, \mathbb{E}\left((Y[1] - Y[0]) - g(X)\right)^2, \tag{2}$$

where $\mathcal{G}$ is the class of all measurable functions $g(\cdot) : \mathcal{X} \to \mathcal{Y}$. Finally, to estimate the causal quantities from the observational data, we make standard identifiability and smoothness assumptions (see Appendix B).

### 3.1 META-LEARNERS FOR CAPOS AND CATE

**Plug-in learners.** A naïve way to estimate CAPOs and CATE is to simply estimate $\hat{\mu}_0^x(x)$ and $\hat{\mu}_1^x(x)$ and 'plug them into' the identification formulas for CAPOs and CATE. For example, an S-learner (S-Net) fits a single model with the treatment as an input, while a T-leaner (T-Net) builds two models

for each treatment (Künzel et al., 2019). Many end-to-end representation learning methods, such as TARNet (Shalit et al., 2017) and BNN without balancing (Johansson et al., 2016), can be seen as variants of the plug-in learner: In the end-to-end fashion, they build a representation of the covariates $\phi = \Phi(x) \in \Phi \subseteq \mathbb{R}^{d_\phi}$ and then use $\phi$ to estimate $\hat{\mu}_a^x(x) = \hat{\mu}_a^\phi(\Phi(x))$ with the S-Net (BNN w/o balancing) or the T-Net (TARNet).

Yet, plug-in learners have several major drawbacks (Morzywolek et al., 2023; Vansteelandt & Morzywołek, 2023). (a) They do not account for the selection bias, namely, that $\hat{\mu}_0^x$ is estimated better for the treated population and $\hat{\mu}_1^x$ for untreated. (b) In the case of CATE estimation, the plug-in learners might additionally fail to address the causal inductive bias that the CATE is a "simpler" function than both CAPOs, as it is impossible to add additional smoothing for the CATE model separately from CAPOs models. (c) It is also unclear how to consistently estimate the CAPOs/CATE depending on the subset of covariates $V \subseteq X$. For example, it is unclear how to estimate representation-level CAPOs, $\xi_a^\phi(\phi) = \mathbb{E}(Y[a] \mid \Phi(X) = \phi)$, and CATE, $\tau^\phi(\phi) = \mathbb{E}(Y[1] - Y[0] \mid \Phi(X) = \phi)$, especially when the representations are constrained.

**Working model & target risks.** To address the shortcomings of plug-in learners, two-stage meta-learners were proposed (see Appendix A.2). They proceed in three steps. ① First, they choose a *target working model class*, $\mathcal{G} = \{g(\cdot) : \mathcal{V} \subseteq \mathcal{X} \to \mathcal{Y}\}$ such as, e.g., neural networks. A target model takes a (possibly confounded) subset $V$ of the original covariates $X$ as an input and outputs the prediction of causal quantities conditioned on $V$, namely CAPOs, $\xi_a^v(v) = \mathbb{E}(Y[a] \mid V = v)$ and CATE, $\tau^v(v) = \mathbb{E}(Y[1] - Y[0] \mid V = v)$.

② Then, two-stage meta-learners formulate one of the *target risks* for $g(v)$, where $v \in \mathcal{V}$. There are multiple choices for choosing a target risk, each with different interpretations and implications for finite-sample two-stage estimation. For example, two usual target risks for CAPOs are based on the MSE (Vansteelandt & Morzywołek, 2023):

$$\mathcal{L}_{Y[a]}(g, \eta) = \mathbb{E}\left(Y[a] - g(V)\right)^2 \qquad \text{and} \qquad \mathcal{L}_{\xi_a}(g, \eta) = \mathbb{E}\left(\mu_a^x(X) - g(V)\right)^2, \qquad (3)$$

where $V \subseteq X$, $\eta = (\mu_a^x, \pi_a^x)$ are nuisance functions (expected covariate-conditional outcomes and covariate propensity score) that influence the target risks. Minimizers of both $\mathcal{L}_{Y[a]}$ and $\mathcal{L}_{\xi_a}$ would be the same if we had access to infinite data for potential outcomes, $Y[a]$, and the ground-truth expected covariate-conditional outcomes, $\mu_a^x$. Yet, the values of both $\mathcal{L}_{Y[a]}$ and $\mathcal{L}_{\xi_a}$ are generally different, which influences finite-sample two-stage learning. At the same, CATE only allows for an MSE target risk, similar to $\mathcal{L}_{\xi_a}$ (Morzywolek et al., 2023):[2]

$$\mathcal{L}_\tau(g, \eta) = \mathbb{E}\left((\mu_1^x(X) - \mu_0^x(X)) - g(V)\right)^2. \qquad (4)$$

Also, for CATE estimation, we can consider an overlap-weighted MSE alternative of $\mathcal{L}_\tau(g)$ (Foster & Syrgkanis, 2023; Morzywolek et al., 2023), namely:

$$\mathcal{L}_{\pi_0 \pi_1 \tau}(g, \eta) = \mathbb{E}\left[\pi_0^x(X)\, \pi_1^x(X)\, ((\mu_1^x(X) - \mu_0^x(X)) - g(V))^2\right]. \qquad (5)$$

Unlike the plug-in learners, the population minimizers of the target risks in Eq. (3) and (4) can recover the representation-level CAPOs/CATE (see Remark 1 of Appendix C).

**Remark 1** (Identifiability of $V$-conditional causal quantities). *The $V$-conditional CAPOs and CATE are identifiable as population minimizers of the target risks from Eq. (3) and (4), respectively, if they are contained in the working model class, i.e., $\xi_a^v \in \mathcal{G}$ and $\tau^v \in \mathcal{G}$.*

③ In the last step, two-stage meta-learners minimize a chosen target risk, $\hat{\mathcal{L}}(g, \hat{\eta})$, which is estimated using observational data and estimated at the first-stage nuisance functions, $\hat{\eta}$. The latest step then yields so-called *Neyman-orthogonal learners* when the target risk is estimated with semi-parametric efficient estimators (Robins & Rotnitzky, 1995; Foster & Syrgkanis, 2023).

**Neyman-orthogonal learners.** Efficient estimation of the target risks introduces the well-known class of Neyman-orthogonal learners. • CAPOs: For example, efficient estimators of MSE target risks for CAPOs yield two DR-learners with the following losses:

$$\hat{\mathcal{L}}_{Y[a]}(g, \hat{\eta}) = \mathbb{P}_n\left\{\frac{\mathbb{1}\{A = a\}}{\hat{\pi}_a^x(X)}\left(Y - g(V)\right)^2 + \left(1 - \frac{\mathbb{1}\{A = a\}}{\hat{\pi}_a^x(X)}\right)\left(\hat{\mu}_a^x(X) - g(V)\right)^2\right\}, \qquad (6)$$

$$\hat{\mathcal{L}}_{\xi_a}(g, \hat{\eta}) = \mathbb{P}_n\left\{\left(\frac{\mathbb{1}\{A = a\}}{\hat{\pi}_a^x(X)}\left(Y - \hat{\mu}_a^x(X)\right) + \hat{\mu}_a^x(X) - g(V)\right)^2\right\}. \qquad (7)$$

---

[2]An analogue to the first target risk of CAPOs, namely, $\mathcal{L}_{Y[1] - Y[0]}(g) = \mathbb{E}\left((Y[1] - Y[0]) - g(V)\right)^2$, contains a counterfactual expression, $Y[1] - Y[0]$, and is, thus, unidentifiable.

The first learner, $\hat{\mathcal{L}}_{Y[a]}(g, \hat{\eta})$, is known as DR-leaner in the style of Foster & Syrgkanis (2023), while the second one, $\hat{\mathcal{L}}_{\xi_a}(g, \hat{\eta})$, is referred to as Kennedy (2023)-style DR-learner. • CATE: Here, an efficient estimator for target MSE, $\mathcal{L}_\tau(g, \eta)$, is the DR-learner in the style of Kennedy (2023); and an efficient estimator for overlap weighted MSE, $\mathcal{L}_{\pi_0 \pi_1 \tau}(g, \eta)$, is the R-learner (Nie & Wager, 2021) with the following loss:

$$\hat{\mathcal{L}}_\tau(g, \hat{\eta}) = \mathbb{P}_n \left\{ \left( \frac{A}{\hat{\pi}_1^x(X)} \big(Y - \hat{\mu}_1^x(X)\big) - \frac{1-A}{\hat{\pi}_0^x(X)} \big(Y - \hat{\mu}_0^x(X)\big) + \hat{\mu}_1^x(X) - \hat{\mu}_0^x(X) - g(V) \right)^2 \right\}, \quad (8)$$

$$\hat{\mathcal{L}}_{\pi_0 \pi_1 \tau}(g, \hat{\eta}) = \mathbb{P}_n \left\{ \Big( \big(Y - \hat{\mu}^x(X)\big) - \big(A - \hat{\pi}_1^x(X)\big) g(V) \Big)^2 \right\}, \quad (9)$$

where $\mu^x(X) = \mathbb{E}(Y \mid X = x) = \pi_1^x(X)\, \mu_1^x(X) + \pi_0^x(X)\, \mu_0^x(X)$.

Apart from addressing the issues of plug-in learners (a)–(c), Neyman-orthogonal learners provide two favorable asymptotical theoretical properties (Foster & Syrgkanis, 2023; Kennedy, 2023): *double robustness and quasi-oracle efficiency*, and, thus, are (in some sense) asymptotically optimal for causal quantities estimation (Balakrishnan et al., 2023). Double robustness states that, if one of the nuisance functions is estimated consistently, then the $V$-conditional CAPOs/CATE are estimated consistently, and quasi-oracle efficiency allows for the minimizer of the target loss with the estimated nuisance functions to be nearly identical to the minimizer of the target loss with the oracle nuisance functions even if the nuisance functions are estimated with slow rates (see Appendix B for the further details). We refer to Remark 2 in Appendix C for a formal statement about double robustness and quasi-oracle efficiency.

**Remark 2** (Double robustness and quasi-oracle efficiency of Neyman-orthogonal learners). *Under mild conditions, the following inequality holds for the estimators of $V$-conditional CAPOs/CATE: the estimated target model, $\hat{g} = \arg\min_{g \in \mathcal{G}} \mathcal{L}(g, \hat{\eta})$, and the ground-truth target model, $g^* = \arg\min_{g \in \mathcal{G}} \mathcal{L}(g, \eta)$:*

$$\|\hat{g} - g^*\|_{L_2}^2 \leq O\big(\mathcal{L}_\diamond(\hat{g}, \hat{\eta}) - \mathcal{L}_\diamond(g^*, \hat{\eta})\big) + R_\diamond^2(\eta, \hat{\eta}), \quad (10)$$

*where $\diamond \in \{Y[a], \xi_a, \tau, \pi_0 \pi_1 \tau\}$, and $R_\diamond^2(\eta, \hat{\eta})$ is a second-order remainder which includes nuisance functions estimation errors of the higher order $(> 2)$. Moreover, DR-learners for CATE and CAPOs obtain the double robustness property.*

## 4 ORTHOGONAL REPRESENTATION LEARNING

**Motivation.** The theory on Neyman-orthogonal learners (Morzywolek et al., 2023; Vansteelandt & Morzywołek, 2023) does not provide a guidance on how to choose $V \subseteq X$. Also, to the best of our knowledge, Neyman-orthogonal learners were not studied through the lens of different representations $\Phi(X)$, chosen in place of $V$. For example, if the representation $\Phi(X)$ itself is learned to be predictive of $\mu_a^x$, as in all the end-to-end representation learning methods, *fitting the target model based on $V = \Phi(X)$ may be beneficial compared to other choices of $V$*. We aim to study this research gap and thus introduce a novel class of Neyman-orthogonal learners called *orthogonal representation learners* (*OR-learners*).

**Overview of our *OR-learners*.** Our *OR-learners* use neural networks to fit a target model $g$ based on the learned representations $\Phi(X)$. They proceed in three stages (see Fig. 2): (0) fitting a representation network, (1) estimating nuisance functions (if necessary), and (2) fitting a target model. At the stage (0), any representation learning method can be performed. Then, at the stage (1), we might need to additionally fit nuisance functions (e. g., when the constrained representations were used at the stage (0) so that $\hat{\mu}_a^\phi$ can be inconsistent wrt. $\hat{\mu}_a^x$). Finally, at the stage (2), we utilize different DR- and R-losses, presented in Sec. 3.1, to fit the target model and, thus, yield a final estimator of CAPOs/CATE.

**Variants.** In the following, we discuss different variants of our *OR-learners* depending on the type of representations they are based: in Sec. 4.1, 4.2, and 4.3 we separately consider unconstrained, constrained invertible and constrained non-invertible representations. For the latter two types of representations, we consider balancing with empirical probability metrics as the main constraint. For each, we present new theoretical results for our *OR-learners*, where we discuss the following questions: (i) How can the learned representation space be interpreted? (ii) Does the representation ensure asymptotic validity in light of the representation-induced confounding bias (RICB)? (iii) How

will our *OR-learners* help in that the target model based on the representation $g(\phi)$ can outperform the original end-to-end representation learning predictor $\hat{\mu}_a^\phi$? and (iv) How can the trained target model be interpreted?

## 4.1 OR-LEARNERS FOR UNCONSTRAINED REPRESENTATIONS

We consider representations $\Phi(X)$ as unconstrained if they are outputs of some fully-connected subnetwork, $FC_\phi$, and the overall output, $\hat{\mu}_a^\phi(\Phi(X))$, aims to minimize a (weighted) MSE loss wrt. to the outcome $Y$ (see stage (0) in Fig. 2). Examples include vanilla representation networks without balancing, e. g., TARNet (Shalit et al., 2017), BNN (Johansson et al., 2016), DragonNet (Shi et al., 2019), CFR-ISW (Hassanpour & Greiner, 2019a), and BWCFR (Assaad et al., 2021).

**(i) Interpretation of the learned representations.** Neural networks can handle increasingly more complicated regression tasks by simply adding more layers. This can be formalized with the notion of (Hölder) smoothness: Each layer induces a new space in which the ground-truth regression function becomes smoother and thus easier to estimate (see Remark 3 in Appendix C).

**Remark 3** (Smoothness of the hidden layers)*. Under mild conditions on the representation network, there exists a hidden layer (marked by $V$) of the network with an increased smoothness.*

In our setting of CAPOs/CATE estimation, we consider $V = \Phi(X)$. Thus, if learned well enough, the representation-subnetwork $FC_\phi$ and the induced representation space $\Phi(\cdot) : \mathcal{X} \to \Phi$ should simplify the task of CAPOs/CATE estimation.

**(ii) Validity wrt. the RICB.** The unconstrained representations $\Phi(X)$ can be also considered asymptotically valid when $d_\phi \geq 2$. As an example of valid representation $\Phi(X)$ with $d_\phi = 2$, we can consider $\{\mu_0^x(X), \mu_1^x(X)\}$ (see Proposition 4 in Appendix C).

**Proposition 4** (Valid unconstrained representation with $d_\phi = 2$)*. The representation $\Phi(X) = \{\mu_0^x(X), \mu_1^x(X)\}$ is valid for CAPOs and CATE.*

These representations can be learned arbitrarily well in the asymptotic regime, given sufficiently deep representation subnetwork, $FC_\phi$, with unconstrained representations (that follows from the universal approximation theorem). Hence, in the case of $d_\phi \geq 2$, the unconstrained representations do not induce the representation-induced confounding bias (RICB). That is, although, in general, $(Y[0], Y[1]) \not\perp\!\!\!\perp A \mid \Phi(X)$, the representation contains all the sufficient information for estimation of $\mu_a^x$, and, hence, the causal quantities can be consistently estimated solely with $\Phi(X)$: $\xi_a^x(x) = \xi_a^\phi(\Phi(x)) = \mu_a^\phi(\Phi(x))$ and $\tau^x(x) = \tau^\phi(\Phi(x)) = \mu_1^\phi(\Phi(x)) - \mu_0^\phi(\Phi(x))$. Additionally, in the absence of constraints and the RICB, the original representation network $\hat{\mu}_a^\phi(\Phi(x))$ can be used as a consistent estimator of $\hat{\mu}_a^x(x)$.

**(iii) How will our *OR-learners* help?** *OR-learners* proceed by using the original representation network as the estimator for $\hat{\mu}_a^x(x)$ and additionally fit a covariate propensity score network $\hat{\pi}_a^x(x)$. Therefore, the second-stage model $g(\phi)$ uses additional propensity information and achieves more efficient estimation. Interestingly, BWCFR without balancing (an inverse propensity of treatment weighted (IPTW) learner) (Assaad et al., 2021) can be seen as a special case of our *OR-learners*. It aims at estimating CAPOs and can achieve Neyman-orthogonality in a single-stage of learning. This happens due to the fact that the target model, $g(x)$, coincides with one of the nuisance functions, $\hat{\mu}_a^x(x)$: In this case, both DR-learner losses from Eq. (6) and (7) simplify to the IPTW-learner loss (= weighted MSE loss of BWCFR):

$$\hat{\mathcal{L}}_{Y[a]}(g \equiv \hat{\mu}_a^x, \hat{\eta}) = \hat{\mathcal{L}}_{\xi_a}(g \equiv \hat{\mu}_a^x, \hat{\eta}) = \mathbb{P}_n \left\{ \frac{\mathbb{1}\{A = a\}}{\hat{\pi}_a^x(X)} \left(Y - \hat{\mu}_a^x(x)\right)^2 \right\}. \tag{11}$$

Notably, the same trick is not possible for CATE estimation, and, therefore, a second-stage model is needed even for BWCFR.

**(iv) Interpretation of the target model.** The fitted target model can be interpreted as some form of a *conditional calibration* of the original representation network. To see that, we can compare our target model, for which $V = \Phi(X)$ holds, with two other alternatives (see stage (0) in Fig. 2): a target model with the input $V = X$ and another target model with the input $V = \{\hat{\mu}_0^x, \hat{\mu}_1^x\}$ (these are also known as prognostic scores; see Appendix A.1). The first option (i.e., $V = X$) suggests fitting the target model completely from scratch and "misses" the opportunity to use learned representations. In addition, the losses of the second-stage model can be highly unstable in low-sample regime (e. g., due to high inverse propensity scores), which hinders the chances of $g(V) = g(X)$ to learn the

Figure 2: **An overview of our *OR-learners*.** Our *OR-learners* proceed in three stages: (0) fitting a representation network, (1) estimation of the nuisance functions (if necessary), and (2) fitting the target models. For the stage (0), we also show different options for the target model input $V$. Depending on the choice of the input $V$, the second-stage model $g(V)$ obtains different interpretations: it either learns a new model from scratch or performs a calibration of the representation network.

representations "from scratch". On the other hand, the second option (i.e., $V = \{\hat{\mu}_0^x, \hat{\mu}_1^x\}$)[3] can only use the outputs of the representation network. For example, the optimal $\hat{g}(\hat{\mu}_0^x(x), \hat{\mu}_1^x(x))$ wrt. the DR-loss in the style of Kennedy (2023) would have the following form:

$$\hat{g}(\hat{\mu}_0^x(x), \hat{\mu}_1^x(x)) = \mathbb{E}\left(\frac{\mathbb{1}\{A = a\}Y}{\hat{\pi}_a^x(X)} \mid \hat{\mu}_a^x(x)\right) + \hat{\mu}_a^x(x)\left[1 - \mathbb{E}\left(\frac{\mathbb{1}\{A = a\}}{\hat{\pi}_a^x(X)} \mid \hat{\mu}_a^x(x)\right)\right]. \quad (12)$$

This implies that $\hat{g}$ performs the average calibration of the original representation network. Therefore, when $V = \Phi(X)$, the target model acts as a *conditional calibration of the original representation network*, namely, a middle ground between full re-training and the calibration on average.

## 4.2 OR-LEARNERS FOR INVERTIBLE REPRESENTATIONS WITH BALANCING

Now, we turn our attention to how our *OR-learners* affect invertible representations, where we enforce additional *balancing with empirical probability metrics*. Balancing aims to minimize some empirical probability metric between treated and untreated distributions of the representations, namely, $\mathrm{dist}(\mathbb{P}(\Phi(X) \mid A = 0), \mathbb{P}(\Phi(X) \mid A = 1))$. To enforce balancing, we use empirical integral probability metrics (IPMs), Wasserstein metric (WM), or maximum mean discrepancy (MMD), as suggested in (Shalit et al., 2017; Johansson et al., 2022) (see definitions in Appendix B). Further, we use normalizing flows (Tabak & Vanden-Eijnden, 2010; Rezende & Mohamed, 2015) for the representation subnetwork $\mathrm{FC}_\phi$ to enforce a strict invertibility. Examples of such networks are CFR (Shalit et al., 2017), CFR-ISW (Hassanpour & Greiner, 2019a), and BWCFR (Assaad et al., 2021),[4] which we call CFRFlow, CFRFlow-ISW, and BWCFRFlow, respectively.

**(i) Interpretation of the learned representations.** As we used a normalizing flow as the representation subnetwork, the transformation $\Phi(\cdot)$ becomes a diffeomorphism. Therefore, it can only non-linearly scale down or up different parts of the original space $\mathcal{X}$. Then, in order to minimize the original MSE loss, the representation network would scale up the parts of space to increase the smoothness of $\mu_a^\phi(\phi)$ (see Remark 3 and Proposition 5 in Appendix C). At the same time, balancing can only scale down regions of the space $\mathcal{X}$ with the lack of overlap (see Proposition 6 in Appendix C).

**Proposition 5** (Smoothness via expanding transformations). *A representation network with a representation $\Phi(X)$ achieves higher Hölder smoothness of $\mu_\phi^a(\cdot)$ by expanding some parts of $\mathcal{X}$.*

**Proposition 6** (Balancing via contracting transformations). *A representation network with a representation $\Phi(X)$ reduces the IPMs, namely, WM and MMD, between the distributions of the representations $\mathbb{P}(\Phi(X) \mid A = 0)$ and $\mathbb{P}(\Phi(X) \mid A = 0)$ by contracting some parts of $\mathcal{X}$.*

Therefore, the final learned representation would combine both scaling up due to effort in smoothing and scaling down due to balancing. If both scaling up and down happen in the different areas of the covariate space, then balancing could be beneficial. On the other hand, if both are happening in the same parts of the space, balancing renders itself useless and any amount of it can only harm the performance of the representation network. This important result allows us to formulate a crucial inductive bias needed for balancing to perform well: *areas with the lack of overlap need to coincide with the areas with low heterogeneity of potential outcomes/treatment effect.*

---

[3]We can also consider $V = \hat{\pi}_1^x$; yet, it yields the same interpretation as $V = \{\hat{\mu}_0^x, \hat{\mu}_1^x\}$.

[4]CFR-ISW and BWCFR additionally implement balancing by re-weighting, using inverse propensities of treatment weights.

Figure 3: **Summary of the insights.** Show are the insights from Sec. 4.2 (left) and 4.3 (right). For both figures, we highlight in yellow boxes how our *OR-learners* (in red) can be beneficial in the comparison with the base representation network (in blue). Specifically, we compare the generalization performances in terms of MSE / precision in estimating heterogeneous effect (PEHE) (lower is better), depending on the strength of balancing, $\alpha$. In both cases, we show the behavior in a finite-sample vs. asymptomatic regime ($n \to \infty$). The plots point to the effectiveness of our *OR-learners* in the asymptotic regime, especially when too much balancing is applied.

**(ii) Validity wrt. the RICB.** Invertible representations can not induce RICB (Melnychuk et al., 2024). However, by scaling up and down different parts of the space $\mathcal{X}$, we can influence the low-sample performance, e. g., the gradient descent depends on the scale of inputs (LeCun et al., 2002).

**(iii) How will our *OR-learners* help?** We build our *OR-learners* similarly to Sec. 4.1, i. e., we used the representation network outputs as the estimators of the nuisance functions, $\hat{\mu}_a^x(x)$. Notably, both CRFFlow-ISW and BWCFRFlow, can be considered Neyman-orthogonal wrt. to the target risks for CAPOs (see the similar argument in Sec. 4.1 (iii)). Our *OR-learners* then will effectively try to "undo" the effect of balancing, as they reintroduce the propensity weighting. Specifically, DR-learners would "re-focus" the target models on the parts of the representation space with the lack of overlap: These regions will have large inverse propensity scores and, thus, the target model will have larger loss there. At the same time, R-learner would be leaning to ignore these regions in its loss.

**(iv) Interpretation of the target model.** As it we describe in (iii), the target model "undoes" the effect of balancing, and, therefore, it slowly loses its interpretation as the conditional calibration model as more balancing is applied. We summarize the benefits of applying our *OR-learners* on top of the invertible representations in Fig. 3 (left).

### 4.3 OR-LEARNERS FOR NON-INVERTIBLE REPRESENTATIONS WITH BALANCING

Finally, we discuss how our *OR-learners* perform based on the non-invertible (general) representations where balancing with empirical probability metrics is enforced. This type of representations were implemented by numerous methods (see the overview in Sec. 2).

**(i) Interpretation of the learned representations.** The learned representations have a similar interpretation as in Sec.4.2 (i). However, the representation network is now not only allowed to scale down or up different parts of the original covariates space, but also to fold it, project it, etc. At the same time, the results of Remark 3, Propositions 5 and 6 still hold in this case. For example, when balancing is applied, non-overlapping parts of the space could be simply folded together (Keup & Helias, 2022) or projected onto some subspace (i. e., transformations with the Lipschitz constant less than one would be applied).

**(ii) Validity wrt. the RICB.** When too much balancing is applied, the representations may (i) lose heterogeneity and (ii) induce the RICB (Melnychuk et al., 2024). That means that (i) no asymptotically consistent estimation based solely on the representations $\Phi(x)$ is possible, e. g., $\xi_a^x(x) \neq \xi_a^\phi(\Phi(x))$; and (ii) the consistent estimation of the representation level causal quantities itself requires the access to the original covariates, i. e., $\xi_a^\phi(\phi) \neq \mu_a^\phi(\phi)$.

**(iii) How will our *OR-learners* help?** Asymptotically, our *OR-learners* will help to remove the RICB so that we can consistently estimate representation level CAPOs and CATE. Yet, they cannot recover the lost heterogeneity and will only estimate causal quantities at $X^y$ level of heterogeneity, where $X^y \subseteq X : X^y \perp\!\!\!\perp A$. Interestingly, in the extreme case of the heterogeneity loss (i. e., when representations are constant, $\Phi(X) = 0$), our *OR-learners* would yield (semi-parametrically) efficient estimators of average potential outcomes (APOs) and average treatment effect (ATE). We refer to Proposition 7 in Appendix C for further details.

**Proposition 7** (Consistent estimation with $\Phi(X) = 0$). *For constant representations $\Phi(X) = 0$, our* OR-learners *yield semi-parametric efficient (augmented inverse propensity of treatment weighted (A-IPTW)) estimators of APOs and ATE / overlap-weighted ATE.*

On the other hand, in the low-sample setting, our *OR-learners* will "undo" the effect of balancing by employing the covariate propensity score. Therefore, our *OR-learners* on the one hand can "undo" the benefit brought by balancing (if there is such a setting), and, on the other, partially fix the damage after applying too much balancing.

**(iv) Interpretation of the target model.** The target model obtains similar interpretation as in Sec. 4.2 (iv). However, in the case of the non-invertible representations with balancing, only $X^y$ level causal quantities can be estimated with the target model. We summarize the benefits of applying our *OR-learners* on top of the non-invertible representations in Fig. 3 (right).

## 5 EXPERIMENTS

**Setup.** We now validate our intuition for *OR-learners* empirically. We follow prior literature (Curth & van der Schaar, 2021b; Melnychuk et al., 2024) and use several (semi-)synthetic datasets where both counterfactual outcomes $Y[0]$ and $Y[1]$ and ground-truth covariate level CAPOs / CATE are available. We perform experiments in three settings, in which we compare the performances of vanilla representation learning methods with our *OR-learners* based on the learned representations. • In **Setting A**, we compare different *OR-learners* based on unconstrained representations. • In **Settings B** and **C**, we show how our *OR-learners* help to improve performance based on invertible and non-invertible representations with balancing, respectively.

**Performance metrics.** We report (i) the out-of-sample root mean squared error (rMSE) and (ii) the root precision in estimating heterogeneous effect (rPEHE) for CAPOs and CATE, respectively. However, as we are interested in how our *OR-learners* improve existing representation learning methods, we report the difference in the performance between the original representation network and our *OR-learners*. Formally, we compute $\Delta_\diamond(\text{rMSE})$ and $\Delta_\diamond(\text{rPEHE})$, where $\diamond \in \{\xi_a, Y[a], \tau, \pi_0\pi_1\tau\}$ is a specific learner for CAPOs or CATE. **Datasets.** We used three standard datasets for benchmarking in causal inference: (1) a fully-synthetic dataset ($d_x = 2$) (Kallus et al., 2019; Melnychuk et al., 2024); (2) the semi-synthetic IHDP dataset ($n = 672 + 75; d_x = 25$) (Hill, 2011; Shalit et al., 2017); (3) a collection of 77 semi-synthetic ACIC 2016 datasets ($n = 4802, d_x = 82$) (Dorie et al., 2019). We refer to Appendix D for further details. **Baselines.** We implemented various state-of-the-art representation learning methods, which act as baselines. We further combine each baseline with our *OR-learners* (see implementation details in Appendix E). Importantly, both the baselines and the combination with our *OR-learners* undergo rigorous hyperparameter tuning, so that the comparison is fair and any performance gain must be attributed to how we integrate a Neyman-orthogonal loss (shown in green number across all tables). The baselines are: **TARNet** (Shalit et al., 2017); several variants of **BNN** (Johansson et al., 2016) (w/ or w/o balancing); several variants of **CFR** (Shalit et al., 2017; Johansson et al., 2022) (w/ balancing, non-/ invertible); several variants of **RCFR** (Johansson et al., 2018; 2022) (different types of balancing); several variants of **CFR-ISW** (Hassanpour & Greiner, 2019a) (w/ or w/o balancing, non-/ invertible); and **BWCFR** (Assaad et al., 2021) (w/ or w/o balancing, non-/invertible).

■ **Setting A.** In Setting A, we want to compare the performance of vanilla representation networks (i.e., TARNet and BNN ($\alpha = 0.0$)) and our *OR-learners* applied on top of the unconstrained representations, where the latter is denoted $V = \Phi(X)$. We compare it with several other variants of our *OR-learners*, where the target network has different inputs: $V = X$ and $V = \{\hat{\mu}_0^x, \hat{\mu}_1^x\}$, yet the same depth of one hidden layer. We also compare with the target model based on the covariates space, but which matches the depth of the original representation network, $V = X^*$ (see Remark 8 in Appendix C for description). Therefore, we provide a fair comparison of our *OR-learners* and other alternative variants of DR/R-learners. **Results.** Table 1 shows the results for the ACIC 2016 dataset collection (we refer to Appendix F for additional results for the synthetic dataset). Therein, our *OR-learners* with $V = \Phi(X)$ achieve superior performance for both CAPOs and CATE. Hence, using the representation $\Phi(X)$ as an input for the target model suggests a good trade-off between full re-training (as it is the case with $V = X^*$ and $V = X$) and a simple averaged calibration, $V = \{\hat{\mu}_0^x, \hat{\mu}_1^x\}$.

■ **Setting B.** Here, we study how our *OR-learners* counteract balancing of the invertible representations. For that, we compare a TARFlow ($\hat{=}$TARNet with a normalizing flow as the representation subnetwork) and other invertible representation networks with varying amounts of balancing $\alpha$: CFR-Flow, CFRFlow-ISW, and BWCFRFlow. For CAPOs estimation, CFRFlow-ISW and BWCFRFlow are already Neyman-orthogonal (see Sec. 4.2) and thus can be considered as special cases of our

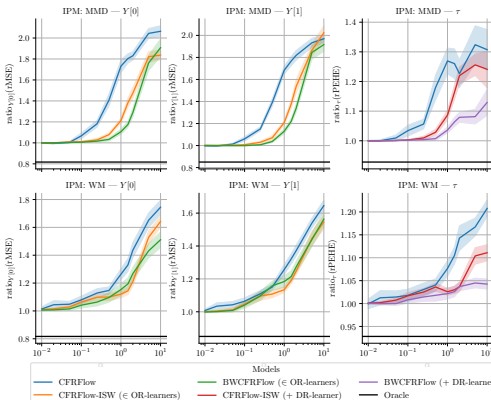

Figure 4: **Results for synthetic experiments in Setting B.** Reported: ratio between the performance of TARFlow (CFRFlow with $\alpha = 0$) and representation networks with varying $\alpha$; mean $\pm$ se over 15 runs. Lower is, thus, better. Here, $n_{\text{train}} = 500, d_\phi = 2$.

Table 1: **Results for 77 semi-synthetic ACIC 2016 experiments in Setting A.** Reported: the percentage of runs, where our *OR-learners* improve over representation networks. Here, $d_\phi = 8$.

| | | $\%_{\varepsilon_0}$ | $\%_{\varepsilon_1}$ | $\%_{Y[0]}$ | $\%_{Y[1]}$ | $\%_\tau$ | $\%_{\pi_0 \pi_1 \tau}$ |
|---|---|---|---|---|---|---|---|
| TARNet | $V = \{\hat{\mu}_0^x, \hat{\mu}_1^x\}$ | 21.30% | 25.71% | 21.04% | 26.49% | 36.88% | 33.51% |
| | $V = X$ | 27.79% | 25.71% | 22.08% | 13.77% | 16.62% | 7.27% |
| | $V = X^*$ | 27.27% | 25.97% | 29.87% | 23.90% | 9.35% | 4.68% |
| | $V = \Phi(X)$ | 60.26% | 58.18% | 68.31% | 67.27% | 70.65% | 69.09% |
| BNN ($\alpha = 0$) | $V = \{\hat{\mu}_0^x, \hat{\mu}_1^x\}$ | 41.04% | 41.30% | 39.22% | 41.56% | 47.27% | 41.56% |
| | $V = X$ | 42.86% | 37.40% | 40.78% | 28.57% | 26.49% | 9.09% |
| | $V = X^*$ | 43.12% | 32.21% | 52.21% | 40.78% | 11.17% | 5.19% |
| | $V = \Phi(X)$ | 63.12% | 73.77% | 81.82% | 67.53% | 87.53% | 84.68% |

Higher = better. Improvement over the baseline in more than 50% of runs marked in green

Table 2: **Results for 77 semi-synthetic ACIC 2016 experiments in Setting C.** Reported: the percentage of runs, where our *OR-learners* improve over representation networks. Here, $d_\phi = 8$.

| | $\%_{\varepsilon_0}$ | $\%_{\varepsilon_1}$ | $\%_{Y[0]}$ | $\%_{Y[1]}$ | $\%_\tau$ | $\%_{\pi_0 \pi_1 \tau}$ |
|---|---|---|---|---|---|---|
| CFR (MMD; $\alpha = 0.1$) | 49.43% | 39.08% | 75.29% | 77.59% | 35.63% | 54.60% |
| CFR (WM; $\alpha = 0.1$) | 58.09% | 53.68% | 77.94% | 76.47% | 45.59% | 53.68% |
| BNN (MMD; $\alpha = 0.1$) | 71.90% | 74.51% | 66.67% | 71.24% | 77.78% | 71.24% |
| BNN (WM; $\alpha = 0.1$) | 81.22% | 74.03% | 75.69% | 76.24% | 82.32% | 80.66% |
| RCFR (MMD; $\alpha = 0.1$) | 65.37% | 49.27% | 73.66% | 78.54% | 52.20% | 62.93% |
| RCFR (WM; $\alpha = 0.1$) | 77.22% | 66.67% | 80.00% | 75.56% | 65.56% | 73.89% |
| CFR-ISW (MMD; $\alpha = 0.1$) | 46.79% | 44.23% | 58.97% | 73.72% | 37.18% | 48.08% |
| CFR-ISW (WM; $\alpha = 0.1$) | 69.68% | 56.13% | 73.55% | 74.84% | 50.32% | 55.48% |
| BWCFR (MMD; $\alpha = 0.1$) | 47.65% | 42.28% | 71.14% | 65.10% | 32.21% | 42.95% |
| BWCFR (WM; $\alpha = 0.1$) | 58.11% | 60.14% | 80.41% | 77.70% | 58.11% | 63.51% |

Higher = better. Improvement over the baseline in more than 50% of runs marked in green

*OR-learners.* For the CATE, we use a second-stage model with the DR-learner. **Results.** The results for Setting B are shown in Fig. 4 (we refer to Appendix F for additional results for the synthetic and IHDP datasets). Therein, CFRFlow-ISW and BWCFRFlow improve the performance of the CFRFlow. The reason is that the synthetic benchmark does not contain instruments and the amount of balancing makes the task of estimating CAPOs/CATE harder.

■ **Setting C.** In the final Setting C, we show how our *OR-learners* "undo" the damage brought by too strict balancing, now including a possible RICB. For this, we use five different representation networks (CFR, BNN, RCFR, CFR-ISW, and BWCFR) as baselines each with two types of balancing and $\alpha = 0.1$: Wasserstein metric (WM) and maximum mean discrepancy (MMD). **Results.** We report the results in Table 2 for the ACIC 2016 dataset collection (we refer to Appendix F for additional results for the synthetic dataset). Here, we filtered only the runs, where balancing representations deteriorated the performance in comparison to the vanilla versions of the representation networks, namely, TARNet for CFR, RCFR, CFR-ISW, and BWCFR; and BNN w/o balancing for BNN. Again, we observe that our *OR-learners* enhance the performance of the representation networks with balancing, even if balancing itself is too restrictive.

**Choice of a target model.** In general, there is no nuisance-free way to do CATE/CAPOs model selection based solely on the observational data (Curth & van der Schaar, 2023). Hence, in the absence of the ground-truth counterfactuals or at least RCT data, one cannot reliably choose among target models with different inputs (e.g., $V = \Phi(X)$ vs. $V = X$) or different hyperparameters (e.g., regularization strength). We can even consider asymptotically-equivalent alternative variants of Neyman-orthogonal learners where constraints are enforced for the second-stage model (see Remark 8 in Appendix C). Yet, our choice of *OR-learners* with $V = \Phi(X)$ is based on (i) a crucial inductive bias that *the high-dimensional covariates lie on some low-dimensional manifold* and (ii) a finite-sample consideration, that the representation network has learned it well in comparison to a second-stage model with an unstable loss (e. g., DR-learner with high inverse propensities).

**Implications.** We discovered that the *inductive bias for balancing is the exact opposite from the regularity conditions of Neyman-orthogonal learners.* In Sec. 4.2, we showed that balancing assumes that the lack of overlap coincides with the lack of potential outcomes/treatment effect heterogeneity (thus, these parts of covariate space will be ignored in the loss of the representation network). On the other hand, Neyman-orthogonal learners do not make such an assumption and consider the areas with the lack of overlap as *uncertain*. For example, the DR-learners would try to infinitely up-weight any observations in those areas (due to extreme inverse propensity weights) and the R-learner would ignore them (assign the weights of zero). Even if the inductive bias (that the lack of overlap implies the lack of heterogeneity) can be assumed, it is still unclear how to choose an optimal amount of balancing on practice (Curth & van der Schaar, 2023). We thus advise against using balancing for representations.

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

## A    EXTENDED RELATED WORK

Our work aims to unify two streams of work, namely, representation learning methods (Sec. A.2) and Neyman-orthogonal learners (Sec. A.1). We review both in the following and then discuss the implications for our work.

### A.1    REPRESENTATION LEARNING FOR ESTIMATING CAUSAL QUANTITIES

Several methods have been previously introduced for *end-to-end* representation learning of CA-POs/CATE (see, in particular, the seminal works by Johansson et al., 2016; Shalit et al., 2017; Johansson et al., 2022). Existing methods fall into three main streams: (1) One can fit an *unconstrained shared representation* to directly estimate both potential outcomes surfaces (e.g., **TARNet** Shalit et al., 2017). (2) Some methods additionally enforce a *balancing constraint based on empirical probability metrics*, so that the distributions of the treated and untreated representations become similar (e.g., **CFR** and **BNN** Johansson et al., 2016; Shalit et al., 2017). Importantly, balancing based on empirical probability metrics is only guaranteed to perform a consistent estimation for *invertible* representations since, otherwise, balancing leads to a *representation-induced confounding bias* (RICB)(Johansson et al., 2019; Melnychuk et al., 2024). Finally, (3) one can additionally perform *balancing by re-weighting* the loss and the distributions of the representations with learnable weights (e.g., **RCFR** Johansson et al., 2022).

Table 3 provides a summary of the main representation learning methods for the estimation of causal quantities. Therein, we showed how different constraints imposed on the representations relate to the consistency of estimation and Neyman-orthogonality of the underlying methods. We highlight several important constrained representations below and discuss the implications for estimating causal quantities.

Table 3: Overview of representation learning methods for CAPOs/CATE estimation. Here, parentheses imply the possibility of an extension.

| Method | Learner type | Constraints | | | Consistency of estimation | Neyman-orthogonality | |
|---|---|---|---|---|---|---|---|
| | | Balancing | Invertibility | Disentanglement | | CAPOs | CATE |
| TARNet (Shalit et al., 2017; Johansson et al., 2022) | PI | – | – | – | ✓ | ✗ | ✗ |
| BNN (Johansson et al., 2016); CFR (Shalit et al., 2017; Johansson et al., 2022); ESCFR (Wang et al., 2024) | PI | IPM | (any) / – | – | ✗ [✓: invertible] | ✗ | ✗ |
| RCFR (Johansson et al., 2018; 2022) | WPI | IPM + LW | (any) / – | | ✗ [✓: invertible] | ✗ | ✗ |
| DACPOL (Atan et al., 2018); CRN (Bica et al., 2020); ABCEI (Du et al., 2021); CT (Melnychuk et al., 2022); MitNet (Guo et al., 2023); BNCDE (Hess et al., 2024) | PI | JSD | – | – | ✗ | ✗ | ✗ |
| SITE (Yao et al., 2018) | PI | LS | MPD | – | ✗ [✓: invertible] | ✗ | ✗ |
| DragonNet (Shi et al., 2019) | PI / (DR) | – | – | – | ✓ | ($✓^{DR_K}$) | ($✓^{DR}$) |
| PM (Schwab et al., 2018); StableCFR (Wu et al., 2023) | WPI | IPM + UVM | – | – | ✓ | ✗ | ✗ |
| CFR-ISW (Hassanpour & Greiner, 2019a) | WPI | IPM + RP | – | – | ✗ | ✗ | ✗ |
| DR-CFR (Hassanpour & Greiner, 2019b); DeR-CFR (Wu et al., 2022) | IPTW | IPM + CP | – | $\Phi = \{\Phi^a, \Phi^\Delta, \Phi^y\}$ | ✓ | ✗ [$✓^{DR}$: IPM = 0] | ✗ |
| DKLITE (Zhang et al., 2020) | PI | CV | RL | – | ✗ [✓: invertible] | ✗ | ✗ |
| BWCFR (Assaad et al., 2021) | IPTW | IPM + CP | – | – | ✓ | ✗ [$✓^{DR}$: IPM = 0] | ✗ |
| SNet (Curth & van der Schaar, 2021b; Chauhan et al., 2023) | DR | – | – | $\Phi = \{\Phi^a, \Phi^\Delta, \Phi^y, \Phi^{\mu_0}, \Phi^{\mu_1}\}$ | ✓ | ($✓^{DR_K}$) | $✓^{DR}$ |
| GWIB (Yang et al., 2024) | PI | MI | – | – | ✗ | ✗ | ✗ |
| ***OR-learners*** (our paper) | DR / R | (any) | NFs / – | (any) | ✓ | $✓^{DR_{FS}}, ✓^{DR_K}$ | $✓^{DR}, ✓^R$ |

*Legend*:
• Learner type: plug-in (PI); weighted plug-in (WPI); inverse propensity of treatment weighted (IPTW); doubly robust (DR); Robinson's / residualized (R)
• Balancing: integral probability metric (IPM); learnable weights (LW); Jensen-Shannon divergence (JSD); local similarity (LS); upsampling via matching (UVM); representation propensity (RP); covariate propensity (CP); counterfactual variance (CV); mutual information (MI)
• Invertibility: middle point distance (MPD); reconstruction loss (RL); normalizing flows (NFs)
• Neyman-orthogonality: DR-learner in the style of Kennedy (2023) ($DR_K$); DR-learner in the style of Foster & Syrgkanis (2023) ($DR_{FS}$)

**Disentanglement.** Shi et al. (2019) proposed to use the shared representation, as in (1) TARNet, to additionally estimate the propensity score. Then, Hassanpour & Greiner (2019b); Wu et al. (2022) suggested disentangling the representation of (1) TARNet or (2) CFR, so that different parts of the disentangled representation can serve to estimate different nuisance functions (potential outcomes surfaces and propensity score). Based on their work, Curth & van der Schaar (2021b) and

Chauhan et al. (2023) further developed a general framework for disentangled representation based on (1) TARNet as a flexible estimator of nuisance functions for different CATE meta-learners.

**Balancing and invertibility.** Following (2) CFR and BNN, several works proposed alternative strategies for *balancing representations with empirical probability metrics*, e. g., based on adversarial learning (Atan et al., 2018; Curth & van der Schaar, 2021a; Du et al., 2021; Melnychuk et al., 2022; Guo et al., 2023); metric learning (Yao et al., 2018); counterfactual variance minimization (Zhang et al., 2020); and empirical mutual information (Yang et al., 2024). To enforce *invertibility* (and, thus, consistency of estimation), several works suggested metric learning heuristics (Yao et al., 2018) or reconstruction loss (Zhang et al., 2020). Other methods, extended *balancing by re-weighting*, as in (3) RCFR, e. g., with weights based on matching (Schwab et al., 2018; Wu et al., 2023); with inverse propensity of treatment weights (IPTW) (Hassanpour & Greiner, 2019a;b; Assaad et al., 2021; Wu et al., 2022).

**Validity of representations for consistent and orthogonal estimation.** As mentioned previously, balancing representations with empirical probability metrics without strictly enforcing invertibility generally leads to *inconsistent estimation based on representations*. This issue was raised as a representation-induced adaptation error (Johansson et al., 2019) in the context of unsupervised domain adaptation and as a representation-induced confounding bias (RICB) (Melnychuk et al., 2024) in the context of estimation of causal quantities. More generally, the RICB can be recognized as a type of runtime confounding (Coston et al., 2020), i. e., when only a subset of covariates is available for the estimation of the causal quantities. Several works offered a solution to circumvent the RICB and achieve consistency, e. g., Assaad et al. (2021) employed IPTW based on original covariates, and Melnychuk et al. (2024) used a sensitivity model to perform a partial identification. However, to the best of our knowledge, no Neyman-orthogonal method was proposed to resolve the RICB (see Fig. 5).

**Note on non-neural representations.** Multiple works also explored the use of non-neural representations for the estimation of causal quantities, also known under the umbrella term of *scores*. Examples include propensity/balancing scores (Rosenbaum & Rubin, 1983; Antonelli et al., 2018), prognostic scores (Hansen, 2008; Huang & Chan, 2017; Luo & Zhu, 2020; Antonelli et al., 2018; D'Amour & Franks, 2021), and deconfounding scores (D'Amour & Franks, 2021). However, we want to highlight that these works focus on different, rather simpler than ours settings:

- *Propensity, balancing, and deconfounding scores* (Rosenbaum & Rubin, 1983) were employed the estimate *average* causal quantities Antonelli et al. (2018); D'Amour & Franks (2021). Examples are average potential outcomes (APOs) and average treatment effect (ATE). This is because they lose information about the heterogeneity of the potential outcomes/treatment effect. In our work, on the other hand, we study a general class of *heterogeneous* causal quantities, namely, representation-conditional CAPOs/CATE.

- *Prognostic scores* (Hansen, 2008) can be used for both averaged (Antonelli et al., 2018; Luo & Zhu, 2020; D'Amour & Franks, 2021) and heterogeneous causal quantities (Huang & Chan, 2017). In (Huang & Chan, 2017; Luo & Zhu, 2020), they are used in the context of a sufficient covariate dimensionality reduction. Yet, these works either (i) make simplifying strong assumptions (Antonelli et al., 2018; Luo & Zhu, 2020; D'Amour & Franks, 2021), so that the prognostic scores coincide with the expected covariate-conditional outcome; or (ii) consider only linear prognostic scores (Huang & Chan, 2017; Luo & Zhu, 2020). To the best of our knowledge, the first practical method for non-linear, learnable representations was proposed by (Johansson et al., 2016; Shalit et al., 2017; Johansson et al., 2022).

Hence, the above-mentioned works operate in much simpler settings and are not relevant baselines for our work.

## A.2 NEYMAN-ORTHOGONAL LEARNERS

**Meta-learners.** Causal quantities can be estimated using model-agnostic methods, so-called *meta-learners* (Künzel et al., 2019). Meta-learners typically combine multiple models to perform two-stage learning, namely, (1) nuisance functions estimation and (2) target model fitting. As such, meta-learners must be instantiated with some machine learning model to perform (1) and (2). Meta-learners have several practical advantages (Morzywolek et al., 2023): (i) they oftentimes offer favorable

theoretical guarantees such as Neyman-orthogonality; (ii) they can address the causal inductive bias that the CATE is "simpler" than CAPOs (Curth & van der Schaar, 2021a), and (iii) the target model obtains a clear interpretation as a projection of the ground-truth CAPOs/CATE on the target model class.

A broad variety of meta-learners have been developed. Notable examples include X- and U-learners (Künzel et al., 2019), R-learner (Nie & Wager, 2021), DR-learner (Kennedy, 2023; Curth et al., 2020), and IVW-learner (Fisher, 2024). Several works extended the theory of targeted maximum likelihood estimation (van der Laan et al., 2011) and proposed Neyman-orthogonal single-stage learners; e. g., EP-learner for CATE (van der Laan et al., 2024) and i-learner for CAPOs (Vansteelandt & Morzywołek, 2023). Furthermore, Curth & van der Schaar (2021b) provided a comparison of meta-learners implemented via neural networks, where disentangled unconstrained representations are used solely to estimate (1) nuisance functions but not as inputs to the (2) target model.

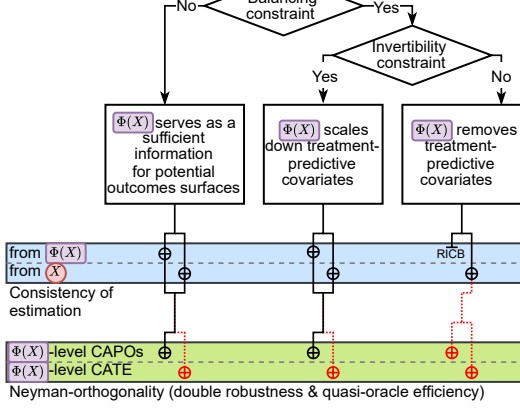

Figure 5: Flow chart of consistency and Neyman-orthogonality for representation learning methods. Our *OR-learners* fill the gaps, marked with red dotted lines.

**Neyman-orthogonality.** Neyman-orthogonality (Foster & Syrgkanis, 2023), or double/debiased machine learning (Chernozhukov et al., 2017), directly extend the idea of semi-parametric efficiency to infinite-dimensional target estimands such as CAPOs and the CATE. Informally, Neyman-orthogonality means that the population loss of the target model is first-order insensitive to the misspecification of the nuisance functions. Examples of Neyman-orthogonal learners are DR-, i-learners for CAPOs (Vansteelandt & Morzywołek, 2023); and DR-, R-, IVW-, EP-learners for CATE (Morzywolek et al., 2023).

**Choice of target models.** Existing works on meta-learners usually build the (2) second-stage target model based on the *original covariates*, for example, the comparative study in (Curth & van der Schaar, 2021b). At the same time, the theory of meta-learners (Morzywolek et al., 2023; Vansteelandt & Morzywołek, 2023) allows for the target model to depend on *any subset of covariates* and to still preserve all the favorable properties (i)-(iii). However, it remained unclear, how different target models relate to each other in terms of (a) performance and (b) interpretation if they are based on different *leaned representations* of covariates. In this paper, we study these questions in detail and introduce *OR-learners*, a novel class of Neyman-orthogonal learners where the target model is based on any representation (with or without constraints).

### A.3 IMPLICATIONS FOR OUR WORK

**Balancing and finite-sample generalization error.** In the original works on balancing representations (Shalit et al., 2017; Johansson et al., 2022), the authors provided finite-sample generalization error bounds for any estimator of CAPOs/CATE based on a factual estimation error and a distributional distance between treated and untreated population. Therein, the authors employed integral probability metrics as the distributional distance. These bounds were further improved with other distributional distances, e. g., counterfactual variance (Zhang et al., 2020), total variation (Csillag et al., 2024), and KL-divergence (Huang et al., 2024). Importantly, the work of the (Shalit et al., 2017; Johansson et al., 2022) suggests that the large distributional distance only *acknowledges the lack of overlap between treated and untreated covariates* (and hence, the hardness of the estimation) and *does not instruct how much balancing needs to be applied*. In our work, we confirm that the optimal amount of balancing is indeed not related to the generalization error bounds.

**Estimation of causal quantities for general-purpose learned representations.** Other constraints may be applied to the representations, e. g., to achieve algorithmic fairness (Zemel et al., 2013; Madras et al., 2018). Although several works combined Neyman-orthogonal learners and fairness constraints,

they were in slightly different from our setting. For example, Kim & Zubizarreta (2023) provided a DR-learner for fair CATE estimation based on the linear combination of the basis functions; and Frauen et al. (2024) built fair representations for policy learning with DR-estimators of policy value. The latter work, nevertheless, can be seen as a special case of our general *OR-learners*.

# B BACKGROUND MATERIALS

In this section, we provide the formal definition of notions such as Neyman-Orthogonality and Hölder smoothness used in Sec 3.

## B.1 ASSUMPTIONS

**Identifiability.** The identification of CAPOs/CATE from observational data requires further assumptions, which are standard in the literature (Rubin, 1974). The reason is that the fundamental problem of causal inference: the counterfactual outcomes, $Y[1-A]$, are never observed, while the potential outcomes are only partially observed, i. e., $Y = A\,Y[1] + (1-A)Y[0]$. Therefore, it is standard to assume (i) *consistency*: if $A = a$, then $Y[a] = Y$; (ii) *overlap*: $\mathbb{P}(0 < \pi_a^x(X) < 1) = 1$; and (iii) *unconfoundedness*: $(Y[0], Y[1]) \perp\!\!\!\perp A \mid X$. Given the assumptions (i)–(iii), both CAPOs and CATE are identifiable from observational data as expected covariate-conditional outcomes, $\xi_a^x(x) = \mu_a^x(x)$, or as the difference of expected covariate-conditional outcomes, $\tau^x(x) = \mu_1^x(x) - \mu_0^x(x)$, respectively.

**Smoothness.** To consistently estimate CAPOs and CATE (e. g., with neural networks), we follow Curth & van der Schaar (2021b); Kennedy (2023) and make regular (Hölder) smoothness assumptions (see Appendix B for the definition). We assume the ground-truth response function $\mu_a^x(\cdot)$ to be $\beta_a$-smooth, the ground-truth propensity score $\pi_a^x(\cdot)$ to be $\gamma$-smooth, and $\tau^x(\cdot)$ to be $\delta$-smooth (for $\beta_a, \gamma, \delta > 0$).

## B.2 NEYMAN-ORTHOGONALITY AND DOUBLE ROBUSTNESS

**Definition 1** (Neyman-orthogonality Foster & Syrgkanis (2023); Morzywolek et al. (2023)). *A risk $\mathcal{L}$, is called Neyman-orthogonal if its pathwise cross-derivative equals to zero, namely,*

$$D_\eta D_g \mathcal{L}(g^*, \eta)[g - g^*, \hat{\eta} - \eta] = 0 \quad \text{for all } g \in \mathcal{G}, \tag{13}$$

*where $D_f F(f)[h] = \frac{\mathrm{d}}{\mathrm{d}t} F(f + th)|_{t=0}$ and $D_f^k F(f)[h_1, \ldots, h_k] = \frac{\partial^k}{\partial t_1 \ldots \partial t_k} F(f + t_1 h_1 + \cdots + t_k h_k)|_{t_1 = \cdots = t_k = 0}$ are pathwise derivatives Foster & Syrgkanis (2023), $g^* = \arg\min_{g \in \mathcal{G}} \mathcal{L}(g, \eta)$, and $\eta$ is the ground-truth nuisance function.*

Informally, this definition means that the risk is first-order insensitive wrt. to the misspecification of the nuisance functions.

**Definition 2** (Double robustness). *An estimator $\hat{g}$ of $g^* = \arg\min_{g \in \mathcal{G}} \mathcal{L}(g, \eta)$ is said to be double robust if, for any estimators $\hat{\mu}_a^x$ and $\hat{\pi}_1^x$ of the nuisance functions $\mu_a^x$ and $\pi_1^x$, it holds that*

$$\|\hat{g} - g^*\|_{L_2}^2 \leq O\big(\mathcal{L}(\hat{g}, \hat{\eta}) - \mathcal{L}(g^*, \hat{\eta})\big) + O_{\mathbb{P}}\big(\|\hat{\pi}_1^x - \pi_1^x\|^2 \|\hat{\mu}_a^x - \mu_a^x\|^2\big), \tag{14}$$

*where $\mathcal{L}(\hat{g}, \hat{\eta}) - \mathcal{L}(g^*, \hat{\eta})$ is the difference between the risks of the estimated target model and the optimal target model where the estimated nuisance functions are used.*

**Definition 3** (Quasi-oracle efficiency). *An estimator $\hat{g}$ of $g^* = \arg\min_{g \in \mathcal{G}} \mathcal{L}(g, \eta)$ is said to be quasi-oracle efficient if the estimators $\hat{\mu}_a^x$ and $\hat{\pi}_1^x$ of the nuisance functions $\mu_a^x$ and $\pi_1^x$ are allowed to have slow rates of convergence, $o(n^{-1/4})$ and the following still holds asymptomatically:*

$$\|\hat{g} - g^*\|_{L_2}^2 \lesssim O\big(\mathcal{L}(\hat{g}, \hat{\eta}) - \mathcal{L}(g^*, \hat{\eta})\big) + o_{\mathbb{P}}(n^{-1/2}), \tag{15}$$

*where $\mathcal{L}(\hat{g}, \hat{\eta}) - \mathcal{L}(g^*, \hat{\eta})$ is the difference between the risks of the estimated target model and the optimal target model where the estimated nuisance functions are used.*

## B.3 HÖLDER SMOOTHNESS

**Definition 4** (Hölder Smoothness). *Let $\beta > 0, C > 0$ and $\mathcal{X} \subseteq \mathbb{R}^{d_x}$. A function $f : \mathcal{X} \to \mathbb{R}$ is said to be $\beta$-Hölder smooth (i.e., belongs to the Hölder class $C^\beta(\mathcal{X})$) if it satisfies the following conditions:*

   *1. $f$ is $\lfloor \beta \rfloor$ times continuously differentiable on $\mathcal{X}$, where $\lfloor \beta \rfloor$ denotes the largest integer less than or equal to $\beta$.*

2. *All partial derivatives of $f$ of order $\lfloor\beta\rfloor$ satisfy the Hölder condition of order $\beta - \lfloor\beta\rfloor$. Specifically, there exists a (Lipschitz) constant $C > 0$ such that for all multi-indices $\alpha$ with $|\alpha| = \lfloor\beta\rfloor$ and for all $x, x' \in \mathcal{X}$,*

$$|D^\alpha f(x) - D^\alpha f(x')| \leq C \|x - x'\|_2^{\beta - \lfloor\beta\rfloor},$$

*where $D^\alpha f$ denotes the partial derivative of $f$ corresponding to the multi-index $\alpha$, and $\|\cdot\|_2$ is the Euclidean norm.*

In our context:

- For each treatment level $a$, the function $\mu_a^x(\cdot)$ is assumed to be $\beta_a$-Hölder smooth with $\beta_a > 0$.

- The propensity score $\pi_a^x(\cdot)$ is assumed to be $\gamma$-Hölder smooth with $\gamma > 0$.

- The conditional average treatment effect function $\tau^x(\cdot)$ is assumed to be $\delta$-Hölder smooth with $\delta > 0$.

### B.4 INTEGRAL PROBABILITY METRICS

Integral probability metrics (IPMs) are a broad class of distances between probability distributions, defined in terms of a family of functions $\mathcal{F}$. Given two probability distributions $\mathbb{P}(Z_1)$ and $\mathbb{P}(Z_2)$ over a domain $\mathcal{Z}$, an IPM measures the maximum difference in expectation over a class of functions $\mathcal{F}$:

$$\text{IPM}(\mathbb{P}(Z_1), \mathbb{P}(Z_2)) = \sup_{f \in \mathcal{F}} |\mathbb{E}(f(Z_1)) - \mathbb{E}(f(Z_2))|.$$

In this framework, $\mathcal{F}$ specifies the allowable ways in which the difference between the distributions can be measured. Depending on the choice of $\mathcal{F}$, different IPMs arise.

**Wasserstein metric (Earth Mover's Distance).** The Wasserstein metric is a specific IPM where the function class $\mathcal{F}$ is the set of 1-Lipschitz functions, which are functions where the absolute difference between outputs is bounded by the absolute difference between inputs:

$$W(\mathbb{P}(Z_1), \mathbb{P}(Z_2)) = \sup_{f \in \mathcal{F}_1} |\mathbb{E}(f(Z_1)) - \mathbb{E}(f(Z_2))|.$$

This metric can be interpreted as the minimum cost required to transport probability mass from one distribution to another, where the cost is proportional to the distance moved.

**Maximum mean discrepancy (MMD).** Another popular example is the Maximum Mean Discrepancy, where the function class $\mathcal{F}$ corresponds to functions in the unit ball of a reproducing kernel Hilbert space (RKHS), $\mathcal{F}_{\text{RKHS}, 1} = \{f \in \mathcal{H} : \|f\|_{\mathcal{H}} \leq 1\}$:

$$\text{MMD}(\mathbb{P}(Z_1), \mathbb{P}(Z_2)) = \sup_{f \in \mathcal{F}_{\text{RKHS}, 1}} |\mathbb{E}(f(Z_1)) - \mathbb{E}(f(Z_2))|.$$

This discrepancy measure is often used in hypothesis testing and in training generative models, particularly when the distributions are defined over high-dimensional data.

## C   THEORETICAL RESULTS

**Remark 1** (Identifiability of $V$-conditional causal quantities). *Assume that the ground-truth $V$-conditional CAPOs and CATE are contained in the working model class, i. e., $\xi_a^v \in \mathcal{G}$ and $\tau^v \in \mathcal{G}$. Then, the $V$-conditional CAPOs/CATE are identifiable as population minimizers of the following target risks:*

$$\xi_a^v(\cdot) = \arg\min_{g \in \mathcal{G}} \mathcal{L}_{Y[a]}(g, \eta) = \arg\min_{g \in \mathcal{G}} \mathcal{L}_{\xi_a}(g, \eta), \tag{16}$$

$$\tau^v(\cdot) = \arg\min_{g \in \mathcal{G}} \mathcal{L}_\tau(g, \eta) \tag{17}$$

*where $\mathcal{L}_{Y[a]}$ and $\mathcal{L}_{\xi_a}$ are given by Eq. (3), and $\mathcal{L}_\tau$ is given by Eq. (4). Furthermore, if the overlap-weighted $V$-conditional CATE, $\tau_{\pi_0\pi_1}^v(v) = \mathbb{E}(\pi_0^x(X)\,\pi_1^x(X)(\mu_1^x(X) - \mu_0^x(X)) \mid V = v)$, is contained in the working model class, i. e., $\tau_{\pi_0\pi_1}^v \in \mathcal{G}$, the overlap-weighted $V$-conditional CATE is identifiable as a population minimizer of target risk of the R-learner:*

$$\tau_{\pi_0\pi_1}^v(\cdot) = \arg\min_{g \in \mathcal{G}} \mathcal{L}_{\pi_0\pi_1\tau}(g, \eta), \tag{18}$$

*where $\mathcal{L}_{\pi_0\pi_1\tau}$ is given by Eq. (5).*

*Proof.* The proof is adapted from (Vansteelandt & Morzywołek, 2023; Morzywolek et al., 2023). First, it is easy to see that $V$-conditional CAPOs and CATE are identifiable, given the ground-truth nuisance functions (e.g., via G-computation formulas):

$$\tau^v(v) = \mathbb{E}(Y[1] - Y[0] \mid V = v) = \xi_1^v(v) - \xi_0^v(v), \tag{19}$$

$$\xi_a^v(v) = \mathbb{E}(Y[a] \mid V = v) \stackrel{(*)}{=} \mathbb{E}(\mathbb{E}(Y[a] \mid X) \mid V = v) \stackrel{\text{Ass. (iii)}}{=} \mathbb{E}(\mathbb{E}(Y[a] \mid X, A = a) \mid V = v) \tag{20}$$

$$\stackrel{\text{Ass. (i)}}{=} \mathbb{E}(\mathbb{E}(Y \mid X, A = a) \mid V = v) = \mathbb{E}(\mu_a^x(X) \mid V = v), \tag{21}$$

where $(*)$ holds due to the law of iterated expectation.

Then, due to the properties of the mean squared error, the last expression is also a population minimizer of the following target risk:

$$\xi_a^v(v) = \mathbb{E}(\mu_a^x(X) \mid V = v) = \arg\min_{g \in \mathcal{G}} \mathbb{E}\big(\mu_a^x(X) - g(V)\big)^2 = \arg\min_{g \in \mathcal{G}} \mathcal{L}_{\xi_a}(g, \eta). \tag{22}$$

For the same reason, $\tau^v(v)$ is a population minimizer of the risk of the DR-learner, i.e., $\mathcal{L}_\tau$; and $\tau_{\pi_0\pi_1}^v(v)$ is a population minimizer of the risk of the R-learner, i.e., $\mathcal{L}_{\pi_0\pi_1\tau}$. Additionally, the risk $\mathcal{L}_{Y[a]}$ has the same population minimizer as $\mathcal{L}_{\xi_a}$:

$$\arg\min_{g \in \mathcal{G}} \mathcal{L}_{Y[a]}(g, \eta) = \arg\min_{g \in \mathcal{G}} \mathbb{E}\,(Y[a] - g(V))^2 \tag{23}$$

$$= \arg\min_{g \in \mathcal{G}} \left[ \mathbb{E}\,(Y[a] - \mu_a^x(X))^2 + 2\mathbb{E}\,(Y[a] - \mu_a^x(X))\,(\mu_a^x(X) - g(V)) + \mathbb{E}\,(\mu_a^x(X) - g(V))^2 \right] \tag{24}$$

$$= \arg\min_{g \in \mathcal{G}} \left[ 2\mathbb{E}\big( (\mu_a^x(X) - g(V))\,\mathbb{E}\,(Y[a] - \mu_a^x(X) \mid X)\big) + \mathbb{E}\,(\mu_a^x(X) - g(V))^2 \right] \tag{25}$$

$$= \arg\min_{g \in \mathcal{G}} \mathbb{E}\,(\mu_a^x(X) - g(V))^2 = \arg\min_{g \in \mathcal{G}} \mathcal{L}_{\xi_a}(g, \eta). \tag{26}$$

$\square$

**Remark 2** (Double robustness and quasi-oracle efficiency of Neyman-orthogonal learners). *Under mild conditions, the following inequality holds for the estimators of $V$-conditional CAPOs/CATE, the estimated target model $\hat{g} = \arg\min_{g \in \mathcal{G}} \mathcal{L}(g, \hat{\eta})$, and the ground-truth target model, $g^* = \arg\min_{g \in \mathcal{G}} \mathcal{L}(g, \eta)$:*

$$\|\hat{g} - g^*\|_{L_2}^2 \le O\big(\mathcal{L}_\diamond(\hat{g}, \hat{\eta}) - \mathcal{L}_\diamond(g^*, \hat{\eta})\big) + R_\diamond^2(\eta, \hat{\eta}), \tag{27}$$

*where $\diamond \in \{Y[a], \xi_a, \tau, \pi_0\pi_1\tau\}$, and $R^2_\diamond(\eta, \hat{\eta})$ is a second-order remainder which includes nuisance functions estimation errors of the higher order. Specifically, $R^2_\diamond(\eta, \hat{\eta})$ are as follows:*

$$R^2_{Y[a]}(\eta, \hat{\eta}) = R^2_{\xi_a}(\eta, \hat{\eta}) = O_\mathbb{P}\big( \|\hat{\mu}^x_a - \mu^x_a\|^2_{L_2} \|\hat{\pi}^x_1 - \pi^x_1\|^2_{L_2} \big), \tag{28}$$

$$R^2_\tau(\eta, \hat{\eta}) = \sum_{a \in \{0,1\}} O_\mathbb{P}\big( \|\hat{\mu}^x_a - \mu^x_a\|^2_{L_2} \|\hat{\pi}^x_1 - \pi^x_1\|^2_{L_2} \big), \tag{29}$$

$$R^2_{\pi_0\pi_1\tau}(\eta, \hat{\eta}) = O_\mathbb{P}\big( \|\hat{\pi}^x_1 - \pi^x_1\|^4_{L_2} \big) + \sum_{a \in \{0,1\}} O_\mathbb{P}\big( \|\hat{\mu}^x_a - \mu^x_a\|^2_{L_2} \|\hat{\pi}^x_1 - \pi^x_1\|^2_{L_2} \big). \tag{30}$$

*Hence, even with slow converging estimators of the nuisance functions, all of the mentioned Neyman-orthogonal learners $\diamond \in \{Y[a], \xi_a, \tau, \pi_0\pi_1\tau\}$ achieve quasi-oracle efficiency (see Definition 15 in Appendix B). Moreover, DR-learners for CATE and CAPOs obtain the double robustness property (see Definition 2 in Appendix B).*

*Proof.* We refer to Theorem 1 of (Morzywolek et al., 2023) and Appendix A of (Vansteelandt & Morzywołek, 2023) for the proofs. □

**Remark 3** (Smoothness of the hidden layers). *Let the learned unconstrained representation network consist of the fixed-width fully-connected layers with locally quadratic activation functions. Then, there exists a hidden layer (marked by $V$) of the representation network with increased Hölder smoothness. That is, the expected $V$-conditional outcome, $\mu^v_a(\cdot) \in \tilde{C}^{\tilde{\beta}_a}(\mathcal{V})$, is Hölder smoother[5] than the original expected covariate-conditional outcome, $\mu^x_a(\cdot) \in C^{\beta_a}(\mathcal{X})$:*

$$\tilde{\beta}_a \leq \beta_a \quad and \quad \tilde{C} \leq C. \tag{31}$$

*Proof.* (informal) We adopt the proof of Lemma 3(d) from (Ohn & Kim, 2019) and Theorem XI.6 from (Elbrächter et al., 2021).

In Lemma 3(d) from (Ohn & Kim, 2019), the authors formulated an important result for *fixed-width fully-connected neural networks with locally quadratic activation functions*. Informally, Lemma A.3(d) constructs an approximation of a Taylor expansion $f_J(x) = \sum_{k=1}^J \frac{(x-1)^k}{k!}$ by using a fixed-width deep neural network. Here, $f_J(x)$ is an example of a generic $\beta = J$ Hölder-smooth function. Then, the approximation of $f_J(x)$ is done by adding $J$ layers where each layer, $j \in 1, \ldots, J$, is only capable of approximating $f_j(x)$ but not $f_{j+1}(x)$.

Theorem XI.6 of (Elbrächter et al., 2021), on the other hand, shows the impossibility of universal approximation with fixed-width fixed-depth neural networks. That means it is always possible to find a $\beta = 2$-smooth function (with the increasing Lipshitz constant, i.e., second-order derivative) that is impossible to approximate with the fixed-width fixed-depth neural networks. Hence, an increase of either width or depth is required.

Therefore, by (Elbrächter et al., 2021), it is impossible to approximate some functions already for $\beta = 2$ with the fixed width and depth. At the same time, the construction of fixed-width deep networks in (Ohn & Kim, 2019) allows for such an estimation by increasing the depth. Notably, with a similar intuition, the theoretical result (namely, more flexibility requires more layers) holds for general classes of fixed-width deep networks (Hanin, 2019; Kidger & Lyons, 2020).

Our proof then follows by contradiction: There should be a hidden layer with larger smoothness since, otherwise, we would not be able to approximate the function solely with the remaining layers. □

**Proposition 4** (Valid unconstrained representation with $d_\phi = 2$). *The representation $\Phi(X) = \{\mu^x_0(X), \mu^x_1(X)\}$ is valid for CAPOs and CATE, namely:*

$$\xi^x_a(x) = \xi^\phi_a(\Phi(x)) = \mu^\phi_a(\Phi(x)) \quad and \quad \tau^x(x) = \tau^\phi(\Phi(x)) = \mu^\phi_1(\Phi(x)) - \mu^\phi_0(\Phi(x)). \tag{32}$$

---

[5]In our paper, we consider the decrease of both $C$ and $\beta$ as smoothing.

*Proof.* We employ properties of conditional expectations:

$$\tau^\phi(\Phi(x)) = \mathbb{E}(Y[1] - Y[0] \mid \Phi(X) = \Phi(x)) \tag{33}$$

$$= \mathbb{E}\big(\mathbb{E}(Y \mid X, A = 1) - \mathbb{E}(Y \mid X, A = 0) \mid \Phi(X) = \Phi(x)\big) \tag{34}$$

$$= \mathbb{E}\big(\mathbb{E}(Y \mid X, A = 1) \mid (\mu_0^x(x), \mu_1^x(x))\big) - \mathbb{E}\big(\mathbb{E}(Y \mid X, A = 0) \mid (\mu_0^x(x), \mu_1^x(x))\big) \tag{35}$$

$$= \mu_1^x(x) - \mu_0^x(x) = \tau^x(x). \tag{36}$$

On the other hand, the following holds:

$$\tau^\phi(\Phi(x)) = \mathbb{E}\big(\mathbb{E}(Y \mid X, A = 1) \mid (\mu_0^x(x), \mu_1^x(x))\big) - \mathbb{E}\big(\mathbb{E}(Y \mid X, A = 0) \mid (\mu_0^x(x), \mu_1^x(x))\big) \tag{37}$$

$$= \mathbb{E}(Y \mid (\mu_0^x(x), \mu_1^x(x)), A = 1) - \mathbb{E}(Y \mid (\mu_0^x(x), \mu_1^x(x)), A = 0) \tag{38}$$

$$= \mu_1^\phi(\Phi(x)) - \mu_0^\phi(\Phi(x)). \tag{39}$$

The derivation of $\xi_a^x(x) = \xi_a^\phi(\Phi(x)) = \mu_a^\phi(\Phi(x))$ follows analogously. $\square$

**Proposition 5** (Smoothness via expanding transformations). *A representation network with a representation $\Phi(X)$ achieves higher Hölder smoothness of $\mu_\phi^a(\cdot)$ by expanding some parts of the space $\mathcal{X}$. That is, for $\mu_x^a(\cdot) \in C^{\beta_a}(\mathcal{X})$ and $\mu_\phi^a(\cdot) \in \tilde{C}^{\beta_a}(\Phi)$ with $\tilde{C} \leq C$, it is necessary that the following holds:*

$$\mathrm{Lip}(\Phi) \geq 1, \tag{40}$$

*where $\mathrm{Lip}(\Phi)$ is a Lipschitz constant of the transformation $\Phi(\cdot)$. In the case of an invertible transformation, we have $\mathrm{Lip}(\Phi) = \sup_{x \in \mathcal{X}} |\det \Phi'(x)|$ and, thus, $\Phi(\cdot)$ expands (scales up) some parts of the space $\mathcal{X}$.*

*Proof.* The proof follows from the properties of the transformation $\Phi(\cdot)$ as a continously-differential function. On the one hand, by the definition of the Hölder smoothness (see Definition 4):

$$\big|D^\alpha \mu_\phi^a(\phi) - D^\alpha \mu_\phi^a(\phi')\big| \leq \tilde{C}\|\phi - \phi'\|_2^{\beta_a - \lfloor \beta_a \rfloor} \quad \text{for } \phi, \phi' \in \Phi \tag{41}$$

$$|D^\alpha \mu_x^a(x) - D^\alpha \mu_x^a(x')| \leq C\|x - x'\|_2^{\beta_a - \lfloor \beta_a \rfloor} \quad \text{for } x, x' \in \mathcal{X}. \tag{42}$$

On the other hand:

$$\|\Phi(x) - \Phi(x')\|_2 \leq \mathrm{Lip}(\Phi)\,\|x - x'\|_2. \tag{43}$$

Therefore, we yield the following inequalities:

$$\big|D^\alpha \mu_\phi^a(\Phi(x)) - D^\alpha \mu_\phi^a(\Phi(x'))\big| \leq \tilde{C}\|\Phi(x) - \Phi(x')\|_2^{\beta_a - \lfloor \beta_a \rfloor} \tag{44}$$

$$\leq \underbrace{\tilde{C}\big(\mathrm{Lip}(\Phi)\big)^{\beta_a - \lfloor \beta_a \rfloor}}_{C} \|x - x'\|_2^{\beta_a - \lfloor \beta_a \rfloor}. \tag{45}$$

Applying the fact that $\tilde{C} \leq C$ finalizes the proof:

$$\tilde{C} \leq \tilde{C}\big(\mathrm{Lip}(\Phi)\big)^{\beta_a - \lfloor \beta_a \rfloor} \implies \mathrm{Lip}(\Phi) \geq 1. \tag{46}$$

$\square$

**Proposition 6** (Balancing via contracting transformations). *A representation network with a representation $\Phi(X)$ reduces the IPMs, namely, WM and MMD (see definitions in Appendix B.4) between the distributions of the representations $\mathbb{P}(\Phi(X) \mid A = 0)$ and $\mathbb{P}(\Phi(X) \mid A = 0)$ by contracting some parts of the space $\mathcal{X}$. That is, to minimize an IPM (either WM or MMD):*

$$\mathrm{IPM}\big(\mathbb{P}(\Phi(X) \mid A = 0), \mathbb{P}(\Phi(X) \mid A = 1)\big) \leq \mathrm{IPM}\big(\mathbb{P}(X \mid A = 0), \mathbb{P}(X \mid A = 1)\big), \tag{47}$$

*it is necessary that the following holds:*

$$\mathrm{Lip}(\Phi) \leq 1, \tag{48}$$

*where $\mathrm{Lip}(\Phi)$ is a Lipschitz constant of the transformation $\Phi(\cdot)$. In the case of an invertible transformation, $\mathrm{Lip}(\Phi) = \sup_{x \in \mathcal{X}} |\det \Phi'(x)|$, and, thus, $\Phi(\cdot)$ scales down some parts of the space $\mathcal{X}$.*

*Proof.* First, we provide the proof for the Wasserstein metric. The Wasserstein metric between the distributions of the representations can be expressed as

$$W\big(\mathbb{P}(\Phi(X) \mid A = 0), \mathbb{P}(\Phi(X) \mid A = 1)\big) \tag{49}$$

$$= \sup_{f \in \mathcal{F}_1} |\mathbb{E}\left(f(\Phi(X)) \mid A = 0\right) - \mathbb{E}\left(f(\Phi(X)) \mid A = 1\right)| \tag{50}$$

$$= \sup_{f \in \mathcal{F}_1} \left| \int_{\mathcal{X}} f(\Phi(x)) \Big(\mathbb{P}(X = x \mid A = 1) - \mathbb{P}(X = x \mid A = 0)\Big) \mathrm{d}x \right| \tag{51}$$

$$= \sup_{\tilde{f} \in \mathcal{F}_K} \left| \int_{\mathcal{X}} \tilde{f}(x) \Big(\mathbb{P}(X = x \mid A = 1) - \mathbb{P}(X = x \mid A = 0)\Big) \mathrm{d}x \right| \tag{52}$$

$$= K \, W\big(\mathbb{P}(X \mid A = 0), \mathbb{P}(X \mid A = 1)\big), \tag{53}$$

where $K$ is a Lipschitz constant of $\Phi(\cdot)$, and the latter equality follows from properties of the Wasserstein metric. Then, we see that the desired inequality in Eq. (47) holds when $K \leq 1$.

Similarly, the inequality from Eq. (47) can be shown for the maximum mean discrepancy by using a Lipschitzness property of a reproducing kernel Hilbert space (RKHS) (see Proposition 3.1 in (Fiedler, 2023)): all functions $f \in \mathcal{F}_{\mathrm{RKHS},1}$ are Lipschitz with the constant 1. Therefore, for a composition of functions $f \circ \Phi$ to be in the RKHS, i.e., $\mathcal{F}_{\mathrm{RKHS},1}$, it is required that $\mathrm{Lip}(\Phi) \leq 1$.

□

**Proposition 7** (Consistent estimation with $\Phi(X) = 0$)**.** *For constant representations $\Phi(X) = 0$, our OR-learners yield semi-parametric efficient (augmented inverse propensity of treatment weighted (A-IPTW)) estimators of APOs and ATE / overlap-weighted ATE. Specifically, if the target model is characterized by an intercept parameter $\theta \in \mathbb{R}$, namely, $g(\cdot) = \theta$, then the minimization of the OR-learners losses yields the following $\hat{\theta}$:*

$$\hat{\theta}_{\xi_a} = \hat{\theta}_{Y[a]} = \mathbb{P}_n \left\{ \frac{\mathbb{1}\{A = a\}}{\hat{\pi}_a^x(X)} \big(Y - \hat{\mu}_a^x(X)\big) + \hat{\mu}_a^x(X) \right\}, \tag{54}$$

$$\hat{\theta}_\tau = \mathbb{P}_n \left\{ \frac{A}{\hat{\pi}_1^x(X)} \big(Y - \hat{\mu}_1^x(X)\big) - \frac{1 - A}{\hat{\pi}_0^x(X)} \big(Y - \hat{\mu}_0^x(X)\big) + \hat{\mu}_1^x(X) - \hat{\mu}_0^x(X) \right\}, \tag{55}$$

$$\hat{\theta}_{\pi_0 \pi_1 \tau} = \mathbb{P}_n \left\{ \frac{1}{\big(A - \hat{\pi}_1^x(X)\big)^2} \frac{\big(Y - \hat{\mu}^x(X)\big)}{\big(A - \hat{\pi}_1^x(X)\big)} \right\} \tag{56}$$

*Proof.* The proof follows from properties of the (weighted) MSE risks. For $\mathbb{E}(Z - \theta)^2$, as in DR-loss in the style of (Kennedy, 2023), the minimum for a constant $\theta \in \mathbb{R}$ is achieved at $\hat{\theta} = \mathbb{E}(Z)$. For $\mathbb{E}(Z_1 - \theta)^2 + \mathbb{E}(Z_2 - \theta)^2$, as in DR-loss in the style of (Foster & Syrgkanis, 2023), the minimum is achieved at $\hat{\theta} = \mathbb{E}(Z_1 + Z_2)$. For the weighted MSE, $\mathbb{E}\big(w(Z)(Z - \theta)^2\big)$, the minimum is achieved for $\hat{\theta} = \frac{\mathbb{E}(w(Z)Z)}{\mathbb{E}(w(Z))}$. □

**Remark 8** (Alternative construction of Neyman-orthogonal learners for constrained representations)**.** *Let alternative learners targeting at the representation-level CAPOs/CATE be defined in the following way. For a working model, $\tilde{\mathcal{G}} = \{g \circ \Phi(\cdot) : \mathcal{X} \to \mathcal{Y}\}$, we aim to minimize the following target risks:*

$$\tilde{\mathcal{L}}_\diamond(g \circ \Phi, \eta) = \mathcal{L}_\diamond(g \circ \Phi, \eta) + \alpha \, \mathrm{dist}(\mathbb{P}(\Phi(X) \mid A = 0), \mathbb{P}(\Phi(X) \mid A = 1)) \tag{57}$$

*wrt. $g \circ \Phi \in \tilde{\mathcal{G}}$, where $\mathcal{L}_\diamond$ is defined in Eq. (3)-(5) for $\diamond \in \{Y[a], \xi_a, \tau, \pi_0 \pi_1 \tau\}$, and $\mathrm{dist}(\cdot, \cdot)$ is a distributional distance, e.g., an IPM. Then, the (1) $\Phi(X)$-conditional CAPOs and CATE identifiable as population minimizers of the target risks from Eq. (57), if they are contained in the $\mathcal{G} = \{g(\cdot) : \Phi \to \mathcal{Y}\}$. Also, (2) the following target losses are Neyman-orthogonal*

$$\hat{\tilde{\mathcal{L}}}_\diamond(g \circ \Phi, \hat{\eta}) = \hat{\mathcal{L}}_\diamond(g \circ \Phi, \hat{\eta}) + \alpha \, \widehat{\mathrm{dist}}(\mathbb{P}(\Phi(X) \mid A = 0), \mathbb{P}(\Phi(X) \mid A = 1)), \tag{58}$$

*where $\mathcal{L}_\diamond$ is defined in Eq. (3)-(5) for $\diamond \in \{Y[a], \xi_a, \tau, \pi_0 \pi_1 \tau\}$. Therefore, these variants of Neyman-orthogonal learners are asymptotically equivalent to our OR-learners.*

*Proof.* The result (1) follows from the properties of joint optimization of Eq. (57) wrt. $g \circ \Phi \in \tilde{\mathcal{G}}$ and Remark 1.

The Neyman-orthogonality of $\hat{\tilde{\mathcal{L}}}_\diamond$ (2) holds, as the balancing constraint, $\widehat{\mathrm{dist}}(\mathbb{P}(\Phi(X) \mid A = 0), \mathbb{P}(\Phi(X) \mid A = 1))$, is insensitive wrt. the misspecification of the nuisance functions, $\pi_a^x$ and $\mu_a^x$. $\qquad\square$

# D DATASET DETAILS

## D.1 SYNTHETIC DATASET

We utilize a synthetic benchmark with hidden confounding as proposed by Kallus et al. (2019), but modify it by incorporating the confounder as the second observed covariate. Specifically, synthetic covariates $X_1$ and $X_2$, along with treatment $A$ and the outcome $Y$, are generated using the following data-generating process:

$$\begin{cases} X_1 \sim \text{Unif}(-2, 2), \\ X_2 \sim N(0, 1), \\ A \sim \text{Bern}\left(\frac{1}{1+\exp(-(0.75\,X_1 - X_2 + 0.5))}\right) \\ Y \sim N\big((2\,A - 1)\,X_1 + A - 2\,\sin(2\,(2\,A - 1)\,X_1 + X_2) - 2\,X_2\,(1 + 0.5\,X_1), 1\big), \end{cases} \tag{59}$$

where $X_1, X_2$ are mutually independent.

## D.2 IHDP DATASET

The Infant Health and Development Program (IHDP) dataset Hill (2011); Shalit et al. (2017) is a widely-used semi-synthetic benchmark for evaluating treatment effect estimation methods. It consists of 100 train/test splits, with $n_{\text{train}} = 672$, $n_{\text{test}} = 75$, and $d_x = 25$. However, this dataset suffers from significant overlap violations, leading to instability in methods that rely on propensity re-weighting Curth & van der Schaar (2021b); Curth et al. (2021).

## D.3 ACIC 2016 DATASET COLLECTION

The covariates for ACIC 2016 are derived from a large-scale study on developmental disorders (Niswander, 1972). The datasets in ACIC 2016 vary in the number of true confounders, the degree of overlap, and the structure of conditional outcome distributions. ACIC 2016 features 77 distinct data-generating mechanisms, each with 100 equal-sized samples ($n = 4802, d_X = 82$) after one-hot encoding the categorical covariates.

# E  Implementation details and hyperparameters

**Implementation.** We implemented our *OR-learners* in PyTorch and Pyro. For better compatibility, the fully-connected subnetworks have one hidden layer with a tuneable number of units. For the normalizing flow subnetworks, we employed residual normalizing flows Chen et al. (2019) that have three hidden layers with a tuneable synchronous number of units. All the networks for our *OR-learners* (see Stages (0)-(2) in Fig. 2) are trained with AdamW (Loshchilov & Hutter, 2019). Each network was trained with $n_{\text{epoch}} = 200$ epochs for the synthetic dataset and $n_{\text{epoch}} = 50$ for the ACIC 2016 dataset collection.

**Hyperparameters.** We performed hyperparameter tuning at all the stages of our *OR-learners* for all the networks based on five-fold cross-validation using the training subset. At each stage, we did a random grid search with respect to different tuning criteria. Table 4 provides all the details on hyperparameters tuning. For reproducibility, we made tuned hyperparameters available in our GitHub.[6]

Table 4: Hyperparameter tuning for baselines and our *OR-learners*.

| Stage | Model | Hyperparameter | Range / Value |
|---|---|---|---|
| **Stage 0** | TARNet BNN CFR BWCFR | Learning rate | 0.001, 0.005, 0.01 |
| | | Minibatch size | 32, 64, 128 |
| | | Weight decay | 0.0, 0.001, 0.01, 0.1 |
| | | Hidden units in $\text{FC}_\phi$ | $R\,d_x$, $1.5\,Rd_x$, $2\,Rd_x$ |
| | | Hidden units in $\text{FC}_a$ | $R\,d_\phi$, $1.5\,Rd_\phi$, $2\,Rd_\phi$ |
| | | Tuning strategy | random grid search with 50 runs |
| | | Tuning criterion | factual MSE loss |
| | | Optimizer | AdamW |
| | CFR-ISW | Representation network learning rate | 0.001, 0.005, 0.01 |
| | | Propensity network learning rate | 0.001, 0.005, 0.01 |
| | | Minibatch size | 32, 64, 128 |
| | | Representation network weight decay | 0.0, 0.001, 0.01, 0.1 |
| | | Propensity network weight decay | 0.0, 0.001, 0.01, 0.1 |
| | | Hidden units in $\text{FC}_\phi$ | $R\,d_x$, $1.5\,Rd_x$, $2\,Rd_x$ |
| | | Hidden units in $\text{FC}_a$ | $R\,d_\phi$, $1.5\,Rd_\phi$, $2\,Rd_\phi$ |
| | | Hidden units in $\text{FC}_{\pi,\phi}$ | $R\,d_\phi$, $1.5\,Rd_\phi$, $2\,Rd_\phi$ |
| | | Tuning strategy | random grid search with 50 runs |
| | | Tuning criterion | factual MSE loss + factual BCE loss |
| | | Optimizer | AdamW |
| | RCFR | Learning rate | 0.001, 0.005, 0.01 |
| | | Minibatch size | 32, 64, 128 |
| | | Weight decay | 0.0, 0.001, 0.01, 0.1 |
| | | Hidden units in $\text{FC}_\phi$ | $R\,d_x$, $1.5\,Rd_x$, $2\,Rd_x$ |
| | | Hidden units in $\text{FC}_a$ | $R\,d_\phi$, $1.5\,Rd_\phi$, $2\,Rd_\phi$ |
| | | Hidden units in $\text{FC}_w$ | $R\,d_\phi$, $1.5\,Rd_\phi$, $2\,Rd_\phi$ |
| | | Tuning strategy | random grid search with 50 runs |
| | | Tuning criterion | factual MSE loss |
| | | Optimizer | AdamW |
| **Stage 1** | Propensity network | Learning rate | 0.001, 0.005, 0.01 |
| | | Minibatch size | 32, 64, 128 |
| | | Weight decay | 0.0, 0.001, 0.01, 0.1 |
| | | Hidden units in $\text{FC}_{\pi,x}$ | $R\,d_x$, $1.5\,Rd_x$, $2\,Rd_x$ |
| | | Tuning strategy | random grid search with 50 runs |
| | | Tuning criterion | factual BCE loss |
| | | Optimizer | AdamW |
| | Outcomes network | Learning rate | 0.001, 0.005, 0.01 |
| | | Minibatch size | 32, 64, 128 |
| | | Hidden units in $\text{FC}_{\text{CNF}}$ | $R\,d_x$, $1.5\,Rd_x$, $2\,Rd_x$ |
| | | Weight decay | 0.0, 0.001, 0.01, 0.1 |
| | | Tuning strategy | random grid search with 50 runs |
| | | Tuning criterion | factual negative log-likelihood loss |
| | | Optimizer | SGD (momentum = 0.9) |
| **Stage 2** | Target network | Learning rate | 0.005 |
| | | Minibatch size | 64 |
| | | EMA of model weights | 0.995 |
| | | Hidden units in $g$ | Hidden units in $\text{FC}_a$ |
| | | Tuning strategy | no tuning |
| | | Optimizer | AdamW |

$R = 2$ (synthetic data), $R = 1$ (IHDP dataset), $R = 0.25$ (ACIC 2016 datasets collection)

---

[6] https://anonymous.4open.science/r/OR-learners.

## F ADDITIONAL EXPERIMENTS

### F.1 SETTING A

Table 5 shows additional results for the synthetic dataset in Setting A. Therein, we observe that our *OR-learners* with $V = \Phi(X)$ are highly effective in comparison to the DR/R-learners based on the original covariates.

Table 5: **Results for synthetic experiments in Setting A.** Reported: improvements of our *OR-learners* over representation networks; mean over 15 runs. Here, $n_{\text{train}} = 500, d_\phi = 2$.

| | | $\Delta_{\xi_0}$ | $\Delta_{\xi_1}$ | $\Delta_{Y[0]}$ | $\Delta_{Y[1]}$ | $\Delta_\tau$ | $\Delta_{\pi_0 \pi_1 \tau}$ |
|---|---|---|---|---|---|---|---|
| TARNet | $V = \{\hat{\mu}_0^x, \hat{\mu}_1^x\}$ | $-0.002$ | $-0.004$ | $-0.002$ | $-0.004$ | $-0.006$ | $-0.009$ |
| | $V = X$ | $+0.064$ | $+0.078$ | $+0.083$ | $+0.059$ | $-0.018$ | $-0.021$ |
| | $V = X^*$ | $+0.015$ | $+0.015$ | $+0.023$ | $+0.004$ | $-0.013$ | $-0.017$ |
| | $V = \Phi(X)$ | $-0.002$ | $-0.004$ | $\pm 0.000$ | $-0.003$ | $-0.011$ | $-0.012$ |
| BNN ($\alpha = 0.0$) | $V = (\hat{\mu}_0^x(X), \hat{\mu}_1^x(X))$ | $-0.006$ | $-0.009$ | $+0.001$ | $-0.009$ | $-0.007$ | $-0.006$ |
| | $V = X$ | $+0.067$ | $+0.045$ | $+0.101$ | $+0.037$ | $-0.020$ | $-0.023$ |
| | $V = X^*$ | $+0.011$ | $-0.005$ | $+0.023$ | $-0.008$ | $-0.010$ | $-0.017$ |
| | $V = \Phi(X)$ | $-0.008$ | $-0.010$ | $-0.002$ | $-0.011$ | $-0.012$ | $-0.012$ |

Lower = better. Improvement over the baseline in green, worsening of the baseline in red

### F.2 SETTING B

Fig. 6 shows the results for the IHDP dataset in Setting B. Interestingly, here balancing in CFRFlow seems to outperform our *OR-learners* for some values of $\alpha$. This is not surprising, as the IHDP dataset contains strong overlap violations and one of the ground-truth potential outcome surfaces is linear $Y[1]$. However, the optimal $\alpha$ are different for both CAPOs and CATE, which renders balancing impractical.

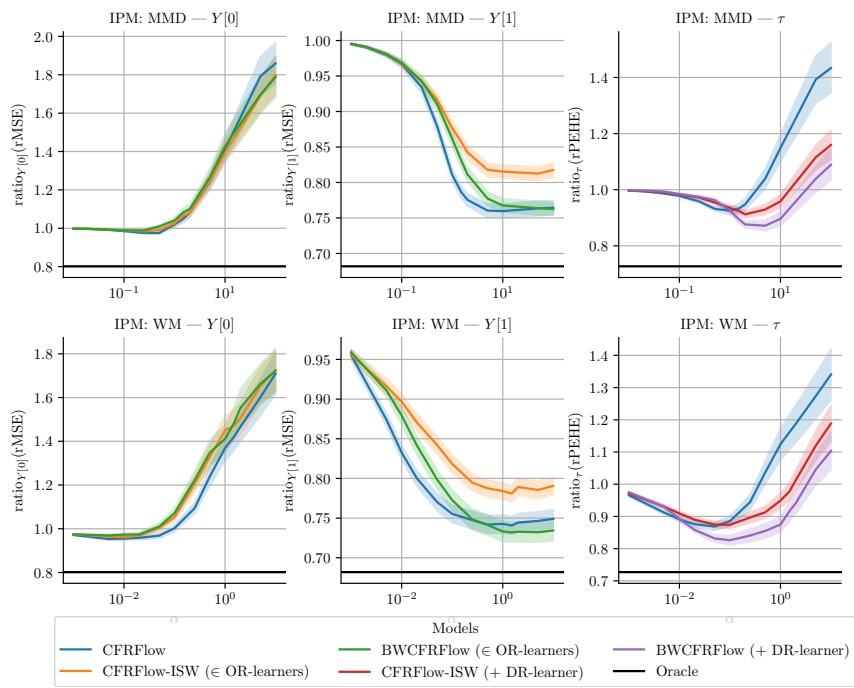

Figure 6: **Results for IHDP experiments in Setting B.** Reported: ratio between the performance of TARFlow (CFRFlow with $\alpha = 0$) and representation networks with varying $\alpha$; mean $\pm$ se over 100 train/test splits.

In Fig. 7, we additionally show how the learned normalizing flows transform the original space $\mathcal{X}$ (the models are the same as in Fig. 4). The rendered transformations match the theoretical results provided in Sec. 4.2. Specifically, TARFlow scales up (expands) the original space so that the regression task

becomes easier in the representation space. At the same time, CRFFlows with different balancing hyperparameters $\alpha$ aim to scale down (contract) the space, thus, achieving better balancing.

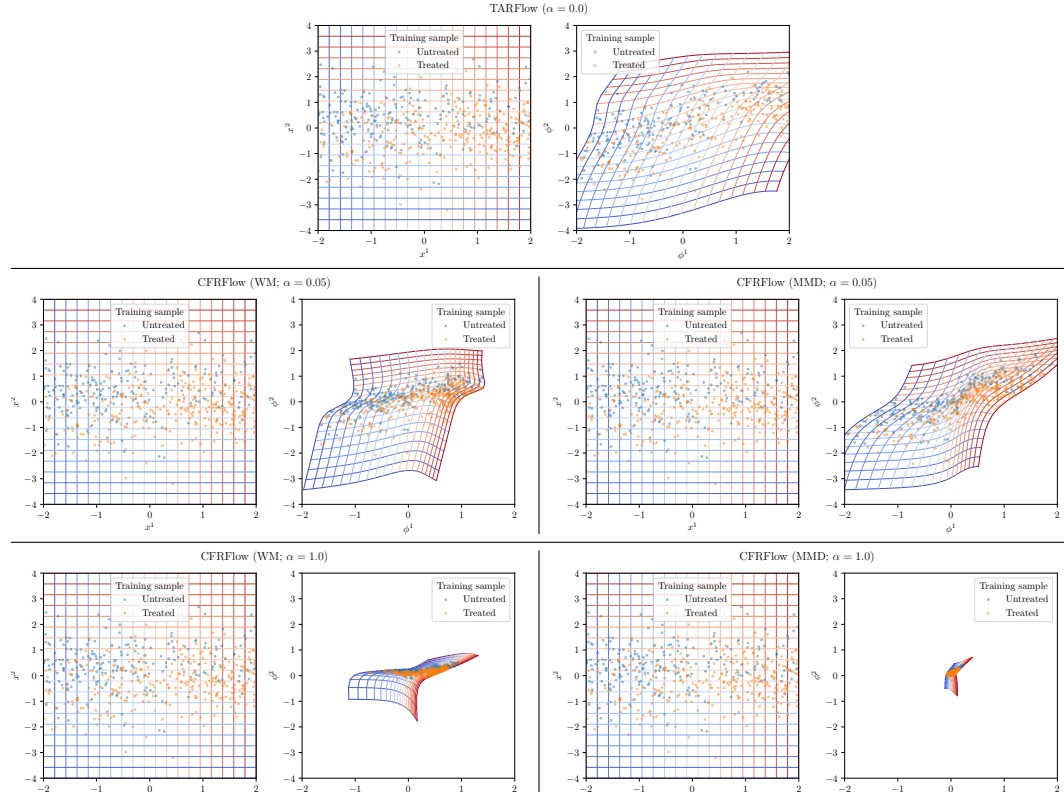

Figure 7: Visualization of the invertible transformations defined by the learned normalizing flow subnetworks for synthetic experiments in Setting B. Here, $n_{\text{train}} = 500, d_\phi = 2$. Specifically, we show how a grid in the original covariate space, $\mathcal{X} \subseteq \mathbb{R}^2$, gets transformed onto the representation space, $\Phi \subseteq \mathbb{R}^2$. We vary the strength of balancing $\alpha \in \{0, 0.05, 1.0\}$ and the IPM $\in \{\text{WM}, \text{MMD}\}$. As suggested by the theory in Sec. 4.2, the covariate space gets scaled up for $\alpha = 0$ and gets scaled up for large values, e.g., $\alpha = 1$.

## F.3  SETTING C

Table 6 shows additional results for the synthetic dataset in setting C. Here, our *OR-learners* improve over the vast majority of the non-invertible representation learning methods where balancing is applied.

Table 6: **Results for synthetic experiments in Setting C.** Reported: improvements of our *OR-learners* over representation networks; mean over 15 runs. Here, $n_{\text{train}} = 500, d_\phi = 2$.

| | $\Delta_{\xi_0}$ | $\Delta_{\xi_1}$ | $\Delta_{Y[0]}$ | $\Delta_{Y[1]}$ | $\Delta_\tau$ | $\Delta_{\pi_0 \pi_1 \tau}$ |
|---|---|---|---|---|---|---|
| CFR (MMD; $\alpha = 0.1$) | $-0.006$ | $-0.009$ | $-0.005$ | $-0.014$ | $-0.011$ | $-0.017$ |
| CFR (WM; $\alpha = 0.1$) | $-0.003$ | $-0.005$ | $-0.006$ | $-0.006$ | $-0.001$ | $-0.005$ |
| BNN (MMD; $\alpha = 0.1$) | $-0.058$ | $-0.011$ | $-0.051$ | $-0.006$ | $-0.048$ | $-0.038$ |
| BNN (WM; $\alpha = 0.1$) | $+0.016$ | $-0.005$ | $-0.013$ | $+0.007$ | $-0.026$ | $-0.026$ |
| RCFR (MMD; $\alpha = 0.1$) | $-0.010$ | $-0.012$ | $-0.032$ | $-0.012$ | $-0.040$ | $-0.028$ |
| RCFR (WM; $\alpha = 0.1$) | $-0.008$ | $-0.003$ | $-0.009$ | $-0.006$ | $-0.019$ | $-0.015$ |
| CFR-ISW (MMD; $\alpha = 0.1$) | $+0.002$ | $-0.002$ | $-0.003$ | $-0.008$ | $+0.001$ | $-0.002$ |
| CFR-ISW (WM; $\alpha = 0.1$) | $+0.001$ | $-0.004$ | $-0.006$ | $-0.003$ | $-0.009$ | $-0.008$ |
| BWCFR (MMD; $\alpha = 0.1$) | $+0.007$ | $-0.005$ | $-0.015$ | $-0.003$ | $-0.015$ | $-0.017$ |
| BWCFR (WM; $\alpha = 0.1$) | $-0.007$ | $-0.008$ | $-0.010$ | $-0.003$ | $-0.010$ | $-0.015$ |

Lower = better. Improvement over the baseline in green, worsening of the baseline in red

