# OpenReview forum: "Orthogonal Representation Learning for Estimating Causal Quantities"
_ICLR.cc/2025/Conference — Submitted to ICLR 2025_

### Official Review · Reviewer_hsvj · 2024-10-27

**Soundness:** 2
**Presentation:** 3
**Contribution:** 2
**Rating:** 8
**Confidence:** 3

**Summary:**

This paper builds on former work on representation learning for treatment effect estimation and pseudo-outcome-based meta-learners satisfying Neyman orthogonality to propose OR-learners, a general meta-learner framework where the pseudo-outcome is learnt not with the original covariates as input features, but with representations that are learnt beforehand using canonical representation learning for treatment effect methods. Intuitions to justify the contribution of the procedure are given, which are further demonstrated in experiments.

**Strengths:**

Originality : It is indeed the first time that I see a contribution on using a two-stage learner, where the two stages are 1) learning a representation to be used as an input for treatment effect estimation, 2) feeding it to pseudo-outcome learning  (with nuisance functions to fit separately) for treatment effect estimation.

Quality&Signifiance: The paper provides an extensive review of the previous literature, and provides an interesting taxonomy of representation learning for treatment effect estimation methods (e.g. without balancing constraints, (non-)invertible with balancing constraints). Experiments seem extensive and able to justify that the method does improve over former baselines.

Clarity : The paper is generally clear in its introductory parts up to Section 3 and in Appendices. I also appreciate the extensive use of figures.

**Weaknesses:**

EDIT Nov 13 : apologies, looks like links did not work! I have re-entered them.

EDIT Dec 1 : edited the score due to author-reviewer discussion.

Originalty&Signifiance : while it is the first work I see learning representations in a "treatment effect estimation friendly" way (i.e.using treatment and/or outcome information together with covariates) to be fed into pseudo-outcome learning, it is not the first method that generally feeds such representations to any treatment effect estimation, even doubly robust methods. More specifically, the submission seems to ignore the extensive and more classical, non-DL literature on such representations, and their use of inputs of treatment effect estimation methods. See for example [propensity scores](https://academic.oup.com/biomet/article/70/1/41/240879), [prognostic scores](https://www.jstor.org/stable/20441477), [sufficient dimension reduction](https://www.tandfonline.com/doi/full/10.1080/07350015.2019.1609974), and [deconfounding scores](https://arxiv.org/pdf/2104.05762). Note that the last reference explicitly feeds a learnt representation into AIPW, a classical doubly-robust method.

Quality&Clarity : besides this former work, what I find is really missing is a mathematical analysis of the proposed method, which is critical as 1) the paper explicitly says that the method "offer[s] favorable properties of Neyman-orthogonality", which are typically demonstrated mathematically, and might be contradicted if the learnt representation is degenerate (e.g. constant), 2) the performance of the method would generally depend on special cases of the representations, e.g. a) it is a constant, as mentioned before, b) it has RICB different from 0, c) it has RICB zero or converging to zero, d) it converges or is equal to a perfect balancing score (predicts treatment assignment) or prognostic score (predicts outcome regression). Instead, the submission uses textual "intuitions" that I find might lack substance, justification or clarity (see questions)

**Questions:**

(Please note that I am very open to increase my score if the above weaknesses and below questions are addressed)

> l.234 : Also, for CATE estimation, we can consider an overlap-weighted MSE alternative of

A reference is missing here.

> l.304-306 : "This can be formalized with the notion of (Holder) smoothness (Ohn & Kim, 2019): Each layer induces a new space in which the ground-truth regression function becomes smoother and thus easier to estimate."

I do not understand exactly where in the reference the claim is justified?

> l. 310-316

Any mathematical or bibliographical justification for why these methods "can be also considered asymptotically valid"? and why specify a dimension of 2 or more?

> l.319-320 : "Therefore, the second-stage model $g(\phi)$ uses additional propensity information and achieves more efficient estimation."

Do you have a mathematical and/or bibliographical (i.e. a reference) justification?

> l.365-367 : "Then, in order to minimize the original MSE loss, the representation network would scale up the parts of space to increase the smoothness of ..."

Do you have a mathematical and/or bibliographical (i.e. a reference) justification?

> l.381-384 : "Our OR-learners then will effectively try to “undo” the effect of balancing, as they reintroduce the propensity weighting. Specifically, DR-learners would “re-focus” the target models on the parts of the representation space with the lack of overlap, while R-learner would ignore them fully"

By balancing, you generally mean minimizing the difference between the distributions of the representation between both groups, right? (Please be specific, as propensity weighting is generally considered balancing!) Also, can you elaborate on this re-focusing will be done? Indeed, with $\Phi(X) = 0$, we have perfectly balanced and overlapping representation, but it cannot be used for any form of estimation!

l> .396-398 : ", but also to fold it, project it, etc. When balancing is applied, non-overlapping parts of the space could be simply folded together or projected onto some subspace, so that they become balanced."

What do you mean exactly by "fold"? That the representation is not injective?

> l.404-405 : "Asymptotically, our OR-learners will help to remove the RICB so that we can consistently estimate representation level CAPOs and CATE."

Justification? Especially if $\Phi(X) = 0$ (which seems to be encapsulated by this section), then the RICB will just never disappear!

> l.407-410 : "On the other hand, in the finite-sample setting, our OR-learners will “undo” the effect of balancing by employing the covariate propensity score. Therefore, our OR-learners on the one hand can “undo” the benefit brought by balancing (if there is such a setting), and, on the other, partially fix the damage after applying too much balancing." :

Tied to the above on l.381-384, but also : can you mathematically or bibliographically justify that this happens in finite-samples?

> l.531-534 : "Informally, balancing assumes that the lack of overlap implies a lack of potential outcomes/treatment effect heterogeneity."

Any mathematical or bibliographical justification?

---

> ### Author Response · Authors · 2024-11-22
> **Response to Reviewer hsvj**
>
> [1 / 5]
>
> We are very grateful for your very comprehensive and actionable review! First of all, we are very happy to hear that you found our idea original and that you appreciated the extensiveness of our related work and experiments. In the following, we respond to each comment and how we improved our paper as a result (see our changes in $\color{blue}{\text{blue}}$ in our **updated PDF**).
>
> ## Response to “Weaknesses”
>
> **Originalty&Signifiance**. Thank you for recognizing the novelty of our work: “The first work I see learning representations … to be fed into pseudo-outcome learning”. Indeed, there are some prior works on non-neural representations [1-4] for causal inference. We thus saw the need to expand our literature review and discuss in greater detail how our work is different and novel (see a **new paragraph in Appendix A.1**).
>
> We are also happy to spell out clearly how our work is different from references [1-4]. Indeed, the papers [1-4] and our work proceed by having the representations as some kind of input for the final stage of the estimation. However, we want to highlight that **the papers [1-4] focus on different, much simpler settings**. Likewise, other papers [5-6] also **focus on different, and rather simple settings**. We make a comparison in the table below. Therein, **we highlight differences across three dimensions**: (i) estimation target, (ii) assumptions, and (iii) linearity of the representation. We give a detailed comparison for each (i)--(iii) below the table.
>
> | **Method**                             | **Estimation target**     | **Additional assumptions**                          | **Linearity of the representation** |
> |----------------------------------------|---------------------------|-----------------------------------------------------|-------------------------------------|
> | [1] (propensity / balancing scores)    | Average                   | **-**                                               | **-**                               |
> | [2] (prognostic score)                 | Average                   | Generalized linear model                            | **-**                               |
> | [3] (prognostic score)                 | Average                   | Generalized linear model                            | Required                            |
> | [4] (deconfounding score)              | Average                   | Generalized linear model, Gaussianity of covariates | Required                            |
> | [5] (prognostic score)                 | Average, **heterogenous** | **-**                                               | Required                            |
> | [6] (prognostic and propensity scores) | Average                   | Generalized  linear model                           | **-**                               |
> | **Our work**                           | **Heterogenous**          | **-**                                               | **-**                               |
>
>
> This is how our work is different from [1-4] and [5-6]:
> - _Estimation target_: The above-mentioned works [1-4, 6] focus on **averaged** causal quantities, like APO, ATE, or ATT. In contrast, our work studies a general class of **heterogeneous** causal quantities: representation-conditional CAPOs/CATE. Therefore, our work is more general than [1-4, 6].
> - _Additional assumptions_. Several of the works require additional, **strong assumptions**. For example, many works on prognostic scores [2-4,6] consider its simplified form ($\Phi(X) = \mathbb{E}(Y[a] \mid X)$), which _requires an additional strong assumption that $Y[a] \mid X$ follows a generalized linear model_. Also, the work of [4] only considers linear deconfounding scores and additionally makes strong parametric assumptions on the data-generating mechanism (Gaussianity of covariates). **In contrast, our work does not make such simplifying assumptions**.
> - _Linear representations_. Arguably, the prognostic scores [2] can be also used as representations for CAPOs/CATE estimation. Yet, the prognostic scores are rather a theoretical concept (unless simplifying assumptions are made [3,4,6]), and only _simple linear_ prognostic scores have been studied so far [3, 5]. To the best of our knowledge, the first practical method for non-linear, learnable representations was proposed by [7-9], while **our work turns [7-9] into Neyman-orthogonal learners**. => The methods in [7-9] (i.e., TARNet, CFR, rCFR) are thus used as baselines in our work.
>
> **Action:** We added a more extensive overview of the literature to our Extended Related Work (see **Appendix A.1 of the updated PDF**). Therein, we spell out clearly how our paper is different from [1-6] and thus how our paper is novel.

---

> ### Author Response · Authors · 2024-11-22
>
> [2 / 5]
>
> **Quality&Clarity**. We are happy to provide an in-depth mathematical analysis in the revised version of the paper. Let us elaborate on the mentioned issues:
>
> 1. To answer your question, we added two new theoretical results,**new Remark 2 and Proposition 7**. In Remark 2, we formalized the favorable theoretical properties (i.e., double robustness and quasi-oracle efficiency) and showed that, even with slow converging estimators of the nuisance functions, our OR-learners consistently and (quasi-oracle) efficiently estimate the main causal quantities: **representation-conditional CAPOs/CATE**. In the case of a degenerate representation (e.g., $\Phi(X) = 0$), our OR-learners still efficiently estimate the causal quantities that are now APOs/ATE (= best constant estimators of CAPOs/CATE). We showed this in our new Proposition 7.
>
>
>    **Action:** We added two formal statements (**new Remark 2 and Proposition 7**), where we explained how our OR-learners achieve consistency and efficiency even in the case of the degenerate representations (e.g., $\Phi(X) = 0$).
>
> 2. We agree, that the performance of the OR-learners as the predictors of the _covariate level CAPOs/CATE_ heavily depends on the choice of the representation (which is expected), as different representations contain different amounts of information about the heterogeneity of the potential outcomes/treatment effect. Yet, in all the mentioned cases, our OR-learners can consistently and (quasi-oracle) efficiently estimate the representation-level causal quantities:
>    - a) See the answer to 1.
>    - b-c) In this case, the representation preserves some (but not all) of the heterogeneity of the potential outcomes/treatment effect. In this case, our OR-learners will adjust for the RICB but still would be limited in their heterogeneity (see a **new Remark 1**). We showcased this with our **new Figure 3**, where we demonstrated finite-sample vs. asymptotic performance of the representation learning methods and our OR-learners.
>
>       **Action:** We added our **new Remark 1** to prove that our OR-learners adjust for the RICB and a **new Figure 3** to describe the performance of our  OR-learners with different amounts of the RICB.
>
>    - d) Very interesting question. In this case, our OR-learners would try to “calibrate” the outputs of the propensity / prognostic score so that the resulting target model would be Neyman-orthogonal (unlike propensity/prognostic scores alone) and, thus, double robust and efficient.
>
>       **Action:** We have extended the discussion on the unconstrained representations in Sec. 4.1 (iv) to include propensity / prognostic scores as possible representations.
>
>    **Action:** We decided to add a paragraph in Sec. 6, where we discuss a choice of the target model’s inputs. Therein, we explained that the good performance of the OR-leaners depends on an important inductive bias: the high-dimensional covariates should lie on a low-dimensional manifold and the learned representations should capture it well. In this case, the target model with the representations $\Phi(X)$ as the input (as in our OR-learners) would be superior over a target model with $X$ as the input (that might struggle to fit $g(X)$ with DR/R-loss due to high-dimensionality of $X$).
>
> ## Response to “Questions”
> 1. **l.234**. Thank you for the suggestion! **Action:** We added additional references to the idea of overlap-weighting the MSE objective for CATE estimation [10-11].
> 2. **l.304-306**. Sorry for the incompleteness of the reference [13]. We decided to be more precise and formulated a new **Remark 3** that states that there exists a hidden layer starting from which the (Hölder) smoothness starts increasing.
>
>    In our proof of the new Remark 3, we refer to the intuition provided in the proof of Theorem 1 in [13] (specifically, the proof of Lemma 3 (d)). Therein, the authors formulated results for _fixed-width neural networks with locally quadratic activation functions_. Specifically, these networks require an increasing number of hidden layers to be able to approximate increasingly non-smoother functions [e.g., Lemma 3 (d) considered Taylor expansion of the square root function with an increasing number of terms]. Similar in intuition theoretical result (more flexibility requires more layers) holds for other types of fixed-width deep networks [14, 15]. Then, our proof follows by contradiction: There should be a hidden layer with larger smoothness since, otherwise, we wouldn’t be able to approximate the function solely with the remaining layers.
>
>    In our context, **Remark 3** holds for the broad class of the representation learning methods. For example, in the experiments, the learned low-dimensional representations, $d_\phi \le d_x$, have fixed dimensionality; and the activation function, ELU, can be considered locally quadratic.
>
>    **Action:** We added a new **Remark 3** and an (informal) proof to our revised PDF, where we explained the reference to [13].

---

> ### Author Response · Authors · 2024-11-22
>
> [3 / 5]
>
> 3.  **l. 310-316**. In this paragraph, we meant the vector of covariate-conditional expected outcomes, $\Phi(X) = (\mu_0^x(X), \mu_1^x(X) )$, as an example of a valid representation with $d_\phi = 2$. We show its validity in our **new Proposition 4 in Appendix C**. Such representations can be learned arbitrarily well in the asymptotic regime, given a sufficiently deep representation subnetwork FC$_\phi$ with unconstrained representations (follows from the universal approximation theorem).
>
>     **Action:** We clarified this statement in the updated version of the PDF (see our **new Proposition 4 in Appendix C**).
>
> 4. **l. 319-320**. Great question! To formalize the efficiency of estimation of the target model $g$, we used the notions of (i) double robustness and (ii) quasi-oracle efficiency (see Sec. 3.2). Also, we added a **new Remark 2** where we precisely stated this efficiency result for our OR-learners. Therein, (i) double robustness implies that, if one of the nuisance functions is estimated consistently, then the projection of the representation-conditional CAPOs/CATE on the working model class, $g^*$, is estimated consistently.  Additionally, (ii) quasi-oracle efficiency means that the minimizer of the target loss of Neyman-orthogonal learners with estimated nuisance functions is asymptotically indistinguishable from the minimizer of the target loss with the ground truth nuisance functions even if the nuisance functions have slow rates of estimation (e.g., $o(n^{-1/4})$. Importantly, quasi-oracle efficiency does not hold generally for non-orthogonal learners (e.g. RA- and IPTW-learners), as they always contain a first-order estimation error of the nuisance functions. The theoretical properties (i) and (ii) thus render our OR-learners asymptotically optimal [18].
>
>    **Action:** We formalized the notation of the target model efficiency in a **new Remark 2** (see the **updated PDF**).
>
> 5. **l.365-367**. To demonstrate why the representation network would scale up the parts of the covariate space, we refer to **new Remark 3 and Proposition 5**. In Remark 3, we showed that to minimize the MSE, a neural network should increase Hölder smoothness with the depth increase (see our answer to Question 2. **l.304-306**). **Proposition 5** then suggests that the  Hölder smoothness increase is only possible when a representation network has a Lipschitz constant larger or equal than one. For the normalizing flows as representation networks,  this implies a determinant of the Jacobian larger than one (which, in turn, implies “scaling up”).
>
>    We also confirmed these results with new experimental evidence. In a **new Figure 7** in Appendix F.2,  we showed how an invertible representation subnetwork scales up the original covariate space when no balancing is applied.
>
>    **Action:** We added new **new Remark 3 and Proposition 5**, where we proved the original claim that the representation network would scale up the parts of the covariate space to minimize the MSE. Also, we experimentally confirmed this result based on the synthetic dataset (see a **new Figure 7** in Appendix F.2).
>
> 6. **l.381-384**. Yes, you are right: By balancing representations in Sec. 4.2 and 4.3, we meant minimizing the distributional difference between the representation of both groups and not balancing with propensity weighting (thank you for this remark).
>
>    We used the term “re-focusing” in the sense that the parts of the covariate space with low overlap would have large inverse propensities in the DR-learner’s loss, thus forcing the target model to focus on them.
>
>    Regarding the degenerate case with  $\Phi(X) = 0$, you raised a very important question (this case can happen only with non-invertible representations). Interestingly, in this case, our OR-learners will lose all the heterogeneity of the potential outcomes/treatment effect **but still can consistently estimate average potential outcomes (APOs) or an average treatment effect (ATE)**. This happens as, in the case of the degenerate representations, the working model class can only fit an intercept and, thus, the objective of the DR-learner coincides with semi-parametric efficient (A-IPTW) estimators of APOs/ATE. We have formalized this interesting result in our **new Proposition 7**.   Note that all the theoretical properties from Remark 2 would still hold (now $V$-conditional CAPOs/CATE became APOs/ATE).
>
>    **Action:** In the revised version of the manuscript, we clarified that balancing (in Sec. 4.2 and 4.3) refers to the minimization of the distributional distance and _not_ to balancing with propensity weighting. Also, we explained what “re-focusing” means for the DR-learner. Additionally, we added our **new Proposition 7**, where we explained that, in the case of the degenerate representations  $\Phi(X) = 0$, our OR-learners yield semi-parametric efficient (A-IPTW) estimators of APOs/ATE.

---

> > ### Author Response · Authors · 2024-11-22
> >
> > [4 / 5]
> >
> > 7. **l.396-398**. By ‘folding’, we meant non-injective transformations, where several values of $x, x’$ are mapped to the same $\phi$ (e.g., after applying ReLU activation [16]). Thus, we do not assume injectivity of the transformation $\Phi(\cdot)$ as the representation is low dimensional ($d_x \ge d_\phi$). We thus added our **new Proposition 6**, which also applies in this case of the non-invertible transformation. Therefore, the non-invertible transformations that enforce balancing are _better characterized by having a Lipschitz constant less than or equal to 1_ (and projection and ‘folding’ are only special cases of those).
> >
> >    **Action:** We added a more general statement with reference to our **new Proposition 6**, which better describes non-invertible transformations that enforce balancing.
> >
> > 8. **l.404-405**. We kindly refer to our answer to 6. **l.381-384*. Therein, we show that in the case of the degenerate representations  $\Phi(X) = 0$, our OR-learners yield semi-parametric efficient (A-IPTW) estimators of APOs/ATE.
> >
> > 9. **l.407-410**. In general, it is very hard to describe the exact finite-sample performance of an estimator of causal quantities without the ground-truth counterfactual data or additional structural assumptions [17, 18] and only loose upper bounds exist (see a discussion in Appendix A.3). It was also shown in [18] that Neyman-orthogonal learners are in many senses optimal but only asymptotically. However, to provide a better outlook of the results in Sec. 4.2 and 4.3., we added a **new Figure 3**, which compares low-sample and asymptotic performances of the base representation learning methods with balancing and the application of our OR-learners on top of them.  Therein, many properties (marked with yellow boxes) in low-sample regimes can possibly happen but are not guaranteed. Yet, in asymptotic regimes, the theoretical results were proved (see a new Remark 2).
> >
> >    **Action:** We added a **new Figure 3** to the revised manuscript to (1) clarify the theoretical results and insights discussed in Sec. 4.2 and 4.3. and to (2) contrast the finite-sample and asymptotic performances of our OR-learners.
> >
> > 10. **l.531-534**. To justify, why balancing requires an implicit assumption that the lack of overlap implies a lack of potential outcomes/treatment effect heterogeneity, we refer to **new Remark 3 and Propositions 5,6**. Therein, we showed that the invertible representation networks with the balancing constraint contain **two different objectives**. On the one hand, the minimization of the MSE forces the network to **scale up** some parts of the input space (so that the ground-truth regression function becomes smoother wrt. to the representation, see Remark 3 & Proposition 5). On the other hand, the minimization of the empirical probability metric (i.e., IPMs), can only be achieved by **scaling down** some parts of the covariate space (see Proposition 6). Thus, we can imagine a data-generating mechanism where the two objectives “collaborate” (= areas of the covariate space with the lack of overlap also lack heterogeneity), and where they “compete” (= lack of overlap coincides with the high heterogeneity). Hence, it is easy to characterize the settings, where balancing either facilitates or harms the overall performance.
> >
> >     This intuition also extends to non-invertible representations with the balancing constraint; yet, the performance here is exacerbated by the possible RICB.
> >
> >     **Action:**  We added the explanation above and the references to **new Remark 3 and Propositions 5,6** to the revised version of our paper.
> >
> > ## References:
> > - [1] Rosenbaum, Paul R., and Donald B. Rubin. "The central role of the propensity score in observational studies for causal effects." Biometrika 70.1 (1983): 41-55.
> > - [2] Hansen, Ben B. "The prognostic analogue of the propensity score." Biometrika 95.2 (2008): 481-488.
> > - [3] Luo, Wei, and Yeying Zhu. "Matching using sufficient dimension reduction for causal inference." Journal of Business & Economic Statistics 38.4 (2020): 888-900.
> > - [4] D'Amour, Alexander, and Alexander Franks. "Deconfounding scores: Feature representations for causal effect estimation with weak overlap." arXiv preprint arXiv:2104.05762 (2021).
> > - [5] Huang, Ming-Yueh, and Kwun Chuen Gary Chan. "Joint sufficient dimension reduction and estimation of conditional and average treatment effects." Biometrika 104.3 (2017): 583-596.
> > - [6] Antonelli, Joseph, et al. "Doubly robust matching estimators for high dimensional confounding adjustment." Biometrics 74.4 (2018): 1171-1179.
> > - [7] Uri Shalit, Fredrik D. Johansson, and David Sontag. Estimating individual treatment effect: Generaliization bounds and algorithms. In International Conference on Machine Learning, 2017.
> > - [8] Fredrik D. Johansson, Uri Shalit, and David Sontag. Learning representations for counterfactual inference. In International Conference on Machine Learning, 2016.

---

> > > ### Author Response · Authors · 2024-11-22
> > >
> > > [5 / 5]
> > >
> > > - [9] Johansson, Fredrik D., et al. "Generalization bounds and representation learning for estimation of potential outcomes and causal effects." Journal of Machine Learning Research 23.166 (2022): 1-50.
> > > - [10] Dylan J. Foster and Vasilis Syrgkanis. Orthogonal statistical learning. The Annals of Statistics, 51(3): 879–908, 2023
> > > - [11] Pawel Morzywolek, Johan Decruyenaere, and Stijn Vansteelandt. On a general class of orthogonal learners for the estimation of heterogeneous treatment effects. arXiv preprint arXiv:2303.12687, 2023.
> > > - [12] Xinkun Nie and Stefan Wager. Quasi-oracle estimation of heterogeneous treatment effects. Biometrika, 108:299–319, 2021.
> > > - [13] Ohn, Ilsang, and Yongdai Kim. "Smooth function approximation by deep neural networks with general activation functions." Entropy 21.7 (2019): 627.
> > > - [14] Hanin, Boris. "Universal function approximation by deep neural nets with bounded width and relu activations." Mathematics 7.10 (2019): 992.
> > > - [15] Kidger, Patrick, and Terry Lyons. "Universal approximation with deep narrow networks." Conference on learning theory. PMLR, 2020.
> > > - [16] Keup, Christian, and Moritz Helias. "Origami in N dimensions: How feed-forward networks manufacture linear separability." arXiv preprint arXiv:2203.11355 (2022).
> > > - [17] Curth A, Van der Schaar M. Nonparametric estimation of heterogeneous treatment effects: From theory to learning algorithms[C]//International Conference on Artificial Intelligence and Statistics. PMLR, 2021: 1810-1818.
> > > - [18] Balakrishnan, Sivaraman, Edward H. Kennedy, and Larry Wasserman. "The fundamental limits of structure-agnostic functional estimation." arXiv preprint arXiv:2305.04116 (2023).

---

> > > > ### Comment · Reviewer_hsvj · 2024-11-24
> > > > **Response to Authors**
> > > >
> > > > Many thanks for the very detailed rebuttal! it already makes a lot of things clearer and more justified.
> > > >
> > > > Here are follow-up questions :
> > > >
> > > > (1) From my understanding, in the end the OR-learners are optimal for the $\Phi(X)$-conditional CAPOs/CATEs, for a fixed $\Phi(X)$.
> > > >
> > > > (a) when you say that Remark 1 proves "that our OR-learners adjust for the RICB", do you mean that they actually just try to find the estimators of the CAPOs/CATEs depending on $\Phi(X)$ with the lowest regression errors, given that the possibly RICB exists i.e. the representation is flawed? Do you have any guarantees on the loss with $V = X$ and the regressor being $g \circ \Phi$, when $g$ comes from minimizing the loss using $V = \Phi(X)$ and $\Phi(X)$ has a given RICB ? This is was what I had in mind as I first understood desirable guarantees, eg doubly robustness or "adjusting for the RICB", i.e. that it is possible to not degrade $X$-conditional CAPO/CATE estimation despite a (not too) bad $\Phi(X)$.
> > > > (b) Is there any way you can find any $\Phi(X)$ minimizing the RICB or reaching a guaranteed RICB?
> > > > (c) Alternatively, can you extend the OR-learners to any adjustment set $V$? Is it really necessary to make it depend on $X$?
> > > >
> > > > (2) To clarify, Prop 6 only holds for Wass and MMD right? Then, please avoid using a generic "IPM" notation, or please specify it is one of Wass or MMD (the latter for any kernel though).
> > > >
> > > > (3) How exactly does Lemma A.3(d) in  (Ohn & Kim, 2019) (page 14 if I am correct) justify that the considered networks "require an increasing number of hidden layers to be able to approximate increasingly non-smoother functions" as you now write in the proof of Remark 3? I struggle to parse their notations!

---

> ### Author Response · Authors · 2024-11-25
>
> [1 / 2]
>
> Thank you for the time devoted to reviewing our paper, we really appreciate your follow-up questions! We have addressed them below (and also in our **freshly updated PDF** where the newest changes are highlighted in $\color{red}{\text{red}}$):
>
> - (1) Your understanding is correct: Our OR-learners are optimal in the sense that they achieve the lowest asymptotic error among the structure-agnostic learners (e.g., RA-/IPTW-learners) of $\Phi(X)$-conditional CAPOs/CATE [1].
>
>    **Action:** We highlighted this fact in the main part of the paper.
>
>
>    > “do you mean that they actually just try to find the estimators of the CAPOs/CATEs depending on $\phi(X)$”
>
>    Yes, you are right. In Remark 1, we established that $\Phi(X)$-conditional CAPOs/CATE are identifiable (i.e., given the infinite amount of the observational data), and, in Remark 2, we provided the asymptotically optimal meta-learners for them.
>
>    > Do you have any guarantees on the loss with $V=X$ and the regressor being $g \circ \Phi$, when $g$ comes from minimizing the loss using $V=\Phi(X)$ and $\Phi(X)$ has a given RICB ?
>
>    This is another very interesting question! If we set $V = X$ and define a second-stage model as $g \circ \Phi$ (with a possibility of additional constraints enforced on $\Phi(X)$), we can get an alternative learner of the representation-level CAPOs/CATE. This new learner would have modified target losses (i.e., Eq. 6/9 + $\alpha$ * balancing constraint) and, still, it would also be Neyman-orthogonal and, thus, asymptotically equivalent to our OR-learners (we show it in our **new Remark 8**). The main reason is that the (balancing) constraint is itself insensitive to the nuisance functions misspecification (i.e., it doesn’t depend on the propensity score or the covariate-conditional expected outcome).
>
>    The natural question arises of how to choose between different variants of Neyman-orthogonal learners: our OR-learners and the newly proposed alternative. Here, there is no universal answer but only **finite-sample considerations**. The same issue exists e.g. for the orthogonal learners of CAPOs [2] where three options are available (=two types of DR-learners and i-leaner). Our main **finite-sample consideration** is that it is that the second-stage model is more optimizationally stable when it has the learned representations $\Phi(X)$ as the inputs in comparison to the suggested model with $g \circ \Phi$, which has to learn the representations $\Phi(X)$ from scratch with a potentially much more unstable loss. The larger instability for the newly proposed learner comes from (a) inverse propensity weights of the DR-loss, (b) additional balancing constraint, and (c) the need to train both $g \circ \Phi$. In our **experiments** (Setting A, $V = X^*$), we verified that **our proposed OR-learners with $V=\Phi(X)$ are indeed more effective than the orthogonal training of the full $g \circ \Phi$**.
>
>    **Action:** We added a **new Remark 8**, where we described an alternative to our OR-learners, as suggested by the reviewer. Furthermore, we added a discussion of how our OR-learners should be preferred in terms of   **finite-sample considerations**.
>
>    > …  it is possible to not degrade $X$-conditional CAPO/CATE estimation despite a (not too) bad $\Phi(X)$.
>
>    Both our OR-leaner and the newly proposed alternative variant would degrade the  $X$-conditional CAPOs/CATE: our OR-learners would use degraded $\Phi(X)$ as inputs and the new alternative would add the balancing term to the loss (thus creating a degrading “bottleneck” wrt. $\Phi(X)$).
>
>    > (b) Is there any way you can find any Φ(X) minimizing the RICB or reaching a guaranteed RICB?
>
>    The RICB can be induced e.g., by balancing constraints. If $\Phi(X)$ aims at minimizing the factual MSE and no balancing is applied (e.g., as in vanilla TARNet, see Sec. 4.1), it doesn’t contain the RICB but only a finite-sample estimation error.
>
>    > (c) Alternatively, can you extend the OR-learners to any adjustment set V? Is it really necessary to make it depend on X?
>
>    Yes, in our OR-learners, $V$ can be any measurable function of $X$ and even a constant (e.g., as suggested in Proposition 7). Thus, the models with $V$ independent of $X$ effectively aim at the APOs / ATE.
>
> - (2) To the best of our knowledge, all the related works on balancing representations chose specific IPMs (WM, MMD, and sometimes, total variation (TV)) but still used the general term to refer to them. However, we agree with you and we will tone down the statement so that it is specific to WM and MMD. Indeed, our Proposition 6 does not hold for TV as the TV is also an f-divergence and, thus, stays constant under the invertible transformations.
>
>    **Action:** We fixed the formulation of Proposition 6 to refer to only WM and MMD.

---

> > ### Author Response · Authors · 2024-11-25
> >
> > [2 / 2]
> >
> > - (3) We want to highlight that Remark 3 is rather an informal statement to provide an intuition about the learned representations (the exact formalism is out of the scope of this paper). However, we are happy to provide explanations of our informal proof.
> >
> >    Informally, Lemma A.3(d) [3] constructs an approximation of a Taylor expansion $f_J(x) = \sum_{k = 1}^{J} \frac{(x-1)^k}{k!}$ by using a fixed-width deep neural network $\theta \in \Theta(L=J, W=14)$ ($L$ and $W$ are the depth and the width, respectively). Here, $f_J(x)$ is an example of a generic $\beta=J$ Hölder-smooth function. Then, the approximation of $f_J(x)$ is done by adding $J$ layers where each layer, $j \in 1, \dots, J$, is only capable of approximating  $f_j(x)$ but not $f_{j+1}(x)$.
> >
> >    To substantiate the main claim that the network “requires an increasing number of hidden layers to be able to approximate increasingly non-smoother functions”, we found additional theoretical results on the impossibility of universal approximation with fixed-width fixed-depth neural networks [4]. Therein, the authors showed in Theorem XI.6 that it is always possible to find a $\beta = 2$-smooth function (with the increasing Lipshitz constant, i.e., second-order derivative) that is impossible to approximate with the fixed-width fixed-depth neural networks. Hence, an increase of either width or depth is required.
> >
> >    Therefore, by [4], it is impossible to approximate some functions already for $\beta = 2$ with the fixed width and depth. On the other hand, the construction of fixed-width deep networks in [3] allows for such an estimation by increasing the depth.
> >
> >    **Action:** We expanded our informal proof of Remark 3 with the explanations above.
> >
> >
> >
> > ### References:
> > - [1] Balakrishnan, Sivaraman, Edward H. Kennedy, and Larry Wasserman. "The fundamental limits of structure-agnostic functional estimation." arXiv preprint arXiv:2305.04116 (2023).
> > - [2] Stijn Vansteelandt and Paweł Morzywołek. Orthogonal prediction of counterfactual outcomes. arXiv preprint arXiv:2311.09423, 2023.
> > - [3] Ohn, Ilsang, and Yongdai Kim. "Smooth function approximation by deep neural networks with general activation functions." Entropy 21.7 (2019): 627.
> > - [4] Elbrächter, Dennis, et al. "Deep neural network approximation theory." IEEE Transactions on Information Theory 67.5 (2021): 2581-2623.

---

> > > ### Comment · Reviewer_hsvj · 2024-11-26
> > > **Proposition 8?**
> > >
> > > Many thanks to authors for this new rebuttal ; I am parsing it but I do not see any Proposition 8, did you actually upload your PDF with Proposition 8?

---

> ### Author Response · Authors · 2024-11-26
>
> Our apology, it should be a **Remark 8**.
>
> We additionally highlighted the newest changes in the PDF with $\color{red}{\text{red color}}$.

---

> > ### Author Response · Authors · 2024-11-27
> >
> > Dear reviewer **hsvj**,
> >
> > We hope we sufficiently addressed all your concerns in our rebuttal. If you have any more questions or concerns, we are happy to answer them as soon as possible. Please let us know if this is the case.
> >
> > Best regards, The authors

---

> > > ### Comment · Reviewer_hsvj · 2024-12-01
> > > **Apologies and score increase**
> > >
> > > Many apologies for answering late, due to personal contraints. All of this looks great, I will increase my score to accept levels. Many thanks for the detailed rebuttals!

---

### Official Review · Reviewer_cqL8 · 2024-11-01

**Soundness:** 3
**Presentation:** 3
**Contribution:** 2
**Rating:** 6
**Confidence:** 4

**Summary:**

The paper focuses on an important problem of estimating causal quantities, including effects and potential outcome mean. This paper proposes to use representations, instead of using original covariates, to learn the causal quantities in the second second stage of DR-learner/R-learner. This paper also provide detailed discussion on the advantages compared with original representation learning methods.

**Strengths:**

1. The paper provides a three-stage causal quantities estimation method, which achieves the DR property.
2. Experimental results verify the effectiveness of the proposed method.

**Weaknesses:**

1. A very key and similar work related to this paper is [1]. My biggest concerns are:
   1. [1] is not well discussed in the main part of the paper. If I understand correctly, the difference between this paper and [1] is that [1] uses original covariates to learn target parameters and this paper uses representations. Could you clarify the advantages compared with [1]?
   2. In experiments, $V=X$ can be seen as an implementation of [1] (please correct me if I misunderstand). However, the conducted experiments only use 1 hidden layer FC, which may explain why the improvement of $V=X$ is much less than $V=\Phi(X)$. It would be fairer and better to use more deep neural networks.
2. It would be better to provide the theoretical improvement that OR-learner brings.
3. It could be better to move Figure 6 to the main body of the paper.

I would be very happy to raise my score if the authors could address my concerns well.

[1] Curth A, Van der Schaar M. Nonparametric estimation of heterogeneous treatment effects: From theory to learning algorithms[C]//International Conference on Artificial Intelligence and Statistics. PMLR, 2021: 1810-1818.

**Questions:**

see weaknesses

---

> ### Author Response · Authors · 2024-11-22
> **Response to Reviewer cqL8**
>
> [1 / 2]
>
> We are grateful for your review! It is great that you found the problem our paper tackles important. Below, we respond to each comment and how we improved our paper as a result (see our changes in $\color{blue}{\text{blue color}}$ in our **updated PDF**).
>
> ## Response to “Weaknesses”
> 1. Thank you for giving us the opportunity to clarify how our work is different from [1]. Despite some similarities with [1], our work can be rather seen as a non-trivial extension of [1]. Let us elaborate on how our work makes important contributions over [1] and how our work is thus novel:
>    1. You are right, the biggest difference between [1] and our work is that we consider the learned representations $\Phi(X)$ as an input to the second-stage model instead of the original covariates $X$. This is non-trivial and thus requires a new derivation. There are two main reasons why such a choice of the target model input might be beneficial for CAPOs/CATE estimation:
>       - **Differences in method:** In Sec. 4.1 (iv), we discuss the unconstrained representations. Therein, the target model with $\Phi(X)$ as inputs provides a middle ground between the full retraining (= with $X$ as inputs as in [1]) and a simple calibration of outputs (= with $\{\hat{\mu}^x_0(X), \hat{\mu}_1^x(X)\}$ as inputs). There, our OR-learners perform a _conditional calibration of the learned representations_ and _enable the theoretical properties_, namely, double robustness & quasi-oracle efficiency. The choice of $\Phi(X)$ as input turns out to be highly effective in the synthetic and semi-synthetic experiments (see Setting A of Sec. 5).
>       - **Differences in setting:** In the context of the constrained representations, the inputs to the target model might be additionally forced to satisfy constraints, e.g., balancing or fairness (if satisfaction of constraints is a crucial requirement, the target model _has to be based on the representations_). Importantly, this setting was **not** studied by [1].    In Sec. 4.2 and 4.3, we specifically focus on balancing representations. We discovered that the orthogonal learners would try to “undo” the effect of balancing: either the benefit of reducing the estimation variance or the damage from too much balancing (=representation-induced confounding bias). These insights constitute our contribution.
>
>       **Action:** We expanded the literature review to spell out more clearly the differences between [1] and our work in the **updated PDF**.
>
>    2. Thank you. You are right: In our experiments, the setting with $V = X$ can be seen as an implementation of [1] with a one-layer neural network as the target model. Yet, in the other setting $V = X^*$, we already implemented exactly what you suggested.  Specifically, we set the target model with the inputs $X$ to match the depth of the original representation network (i.e., the depth of three hidden layers). As seen in Tables 1 and 2, the results $V = X$ and $V = X^*$ are comparable but inferior in comparison with $V = \Phi(X)$, suggesting the effectiveness of our OR-learners.
>
>       **Action:** We provided a clarification in the **updated PDF** that the special case with $ V = X^*$ provides a fair comparison of different target models in terms of the network depths. In numerous experiments, we demonstrated that our OR-learners (which use the learned representations and one hidden layer target model) should be preferred over the full retraining of the target model (which coincides in depth with the original representation network).

---

> > ### Author Response · Authors · 2024-11-22
> >
> > [2 / 2]
> >
> > 2. We are happy to formalize the theoretical improvements that our OR-learners offer.
> >    - **OR-learners vs. end-to-end representation learning**. In comparison with end-to-end representation learning methods, our OR-learners have three main theoretical advantages: (a) they allow to efficiently address selection bias (without inducing representation-induced confounding bias like balancing does), (b) they tackle an inductive bias that CATE is simpler than both of the CAPOs, and (c) they allow to perform unbiased estimation of the CAPOs/CATE based on constrained (thus, potentially confounded) representations. We understand that we should have highlighted theoretical improvements as formal statements, and we thus improved our revised paper as follows. We first formalized (a) as quasi-oracle efficiency and thus added a **new Remark 2** in Sec. 3.2 and Appendix C. We formalized (c) as identifiability of representation-level causal quantities and thus added a **new Remark 1** in Sec. 3.2 and Appendix C.
> >    - **OR-learners with $V = \Phi(X)$ vs. R/DR-learners with $V = X$**. In general, there is no nuisance-free way to do CATE/CAPOs model selection based solely on the observational data [2]. That is, in the absence of the ground-truth counterfactuals or at least RCT data, we can not reliably choose among target models with different inputs (e.g., $V = \Phi(X)$ vs. $V = X$),  or different hyperparameters (e.g., regularization strength). Therefore, our choice of  OR-learners with $V = \Phi(X)$ is based on _the inductive bias that the high-dimensional covariates lie on some low-dimensional manifold_ (which can be learned with the end-to-end representation learning methods _but not the target model based on the original covariates_). This inductive bias is **realistic** in many scenarios: In numerous synthetic and semi-synthetic experiments, especially with high-dimensional covariates (e.g., ACIC-2016 datasets), we demonstrated that our OR-learners are highly effective in comparison with R/DR-learners with $V = X$.
> >
> >    **Action:** We added the information on how our OR-learners improve over existing methods to the **revised PDF**. For this, we added several formal statements (see new **Remarks 1 and 2**), and we added a new discussion about inductive biases (see our new materials in Sec. 6).
> >
> > 3. Great idea! **Action:** We have moved Figure 6 (=new Figure 2) to the main part of the paper.
> >
> >
> > ## References:
> > - [1] Curth A, van der Schaar M. Nonparametric estimation of heterogeneous treatment effects: From theory to learning algorithms[C]//International Conference on Artificial Intelligence and Statistics. PMLR, 2021: 1810-1818.
> > - [2] Curth, Alicia, and Mihaela van der Schaar. "In search of insights, not magic bullets: Towards demystification of the model selection dilemma in heterogeneous treatment effect estimation." International Conference on Machine Learning. PMLR, 2023.

---

> > > ### Comment · Reviewer_cqL8 · 2024-11-25
> > > **Thanks for the rebuttal**
> > >
> > > I appreciate the authors' extremely detailed responses, which have already addressed the majority of my concerns. I will think further about adjusting the score.

---

> > > > ### Author Response · Authors · 2024-12-01
> > > >
> > > > Thank you for your thoughtful feedback! If there are any remaining areas where we can improve our paper, we would be grateful for your guidance. If you feel our responses have addressed your concerns satisfactorily, we would kindly appreciate it if you could consider increasing the score.

---

> > > > > ### Comment · Reviewer_cqL8 · 2024-12-01
> > > > >
> > > > > I have read all of the reviewers' comments and authors' responses, based on which, I will update my score from 5 to 6.

---

### Official Review · Reviewer_v7uW · 2024-11-03

**Soundness:** 2
**Presentation:** 1
**Contribution:** 2
**Rating:** 3
**Confidence:** 3

**Summary:**

The manuscript introduces a new approach to estimating causal quantities by learned representation. With a carefully chosen target risk, a Neyman-orthogonal learner is proposed. Although claiming the theoretical guarantees, the authors do not provide any theoretical analysis. In addition, there is no foundation for some claims (see details in the questions). Lastly, there is still a large room for improvement in the writing to achieve a publication level.

**Strengths:**

The authors raise an interesting question, although the solution can still be improved.

**Weaknesses:**

The author may want to present some evidence for some claims in the manuscript. In addition, combining existing ideas may not be interesting enough. The writing can be much improved.

**Questions:**

1) The manuscript claims that any representation can work. However, proving $Y(0), Y(1)\perp Z|\Phi(X)$ is not easy, and the conditional independence does not hold for any representation. With this condition, the downstream analysis is biased.

2) Formula (10) does not include $g$

3) What is the difference between $V$ and $\Phi(X)$?

4) Estimating nuisance functions can be difficult sometimes. How does the accuracy in estimating nuisance functions affect the final estimation?

---

> ### Author Response · Authors · 2024-11-22
> **Response to Reviewer v7uW**
>
> [1 / 2]
>
> Thank you for your review! We appreciate that you found the topic of our paper interesting. We hope we can address your concerns in the comment below and the revised version of the paper (see our changes in $\color{blue}{\text{blue color}}$ in our **updated PDF**).
>
> ## Response to “Summary”
>
> Thank you for the feedback! We took your suggestion to heart and are happy to provide more theoretical analyses, formal statements, and proofs in the revised version of the paper. In particular, we added the following:
> - We showed that the representation-conditional CAPOs/CATE are identifiable, given the covariate-conditional nuisance functions, and therefore, our OR-learners can adjust for the representation-induced confounding bias (RICB) (see _Remark 1_).
> - In a new _Remark 2_, we formalized the favourable theoretical properties guaranteed by our OR-learners due to their Neyman-orthogonality (namely, double robustness and quasi-oracle efficiency).
> - We demonstrated, that under mild conditions on the representation network, the minimization of the MSE loss requires an increase of (Hölder) smoothness  (see _Remark 3_).
> - We added the proof, that a representation with $d_\phi \le 2$ is sufficient not to induce the RICB (see our new  _Proposition 4_).
> - In our new _Proposition 5_, we proved that, in order for the representation network to increase smoothness, it needs to have a Lipschitz constant of larger or equal than 1.  At the same time, in our new _Proposition 6_, we demonstrated that to achieve balancing, the converse needs to happen (namely the Lipshitz constant needs to be less or equal than 1).
> - Finally, in our new  _Proposition 7_, we showed that in the extreme case of a constant representation, our OR-learners yield efficient estimators of APOs/ATE.
>
> **Action:** We added the above statements as new remarks and propositions to our revised paper. We also added the corresponding proofs to a **new Appendix C**.
>
> ## Response to “Weaknesses”
>
> > The author may want to present some evidence for some claims in the manuscript.
>
> We are happy to provide the proofs and explanations for the claims in Sec. 3 and 4 (see the response to “Summary”).
>
> **Action:** We added the above statements as new remarks and propositions to our revised paper. We also added the corresponding proofs to a **new Appendix C**.
>
> > In addition, combining existing ideas may not be interesting enough. The writing can be much improved.
>
> Our paper is located at the intersection of two literature streams (see Appendix A): end-to-end representation learning methods and Neyman-orthogonal learners. Yet, to the best of our knowledge, both streams were developing almost independently and **no work discussed a unification of both streams, which is our novelty**. As such, the combination of both theories requires an **extensive, non-trivial discussion**, which we provide in Sec. 4. Importantly, the insights provided by our work might be useful for applied CAPOs/CATE estimation (e.g., discovered connections between balancing representations and meta-learners). That is the researchers _should not be burdened with implementing and evaluating dozens of seemingly similar representation learning methods_ (see Table 5) but rather **put their effort into choosing a meta-learner** (R-/DR-learner).
>
> **Action:** We added more explanations about the novelty of our work and why the combination of two main streams of work (representation learning methods and Neyman-orthogonal learners) is non-trivial and requires extensive discussion.
>
>
> ## Responses to “Questions”
>
> 1. Thank you for the question. We do not require conditional independence (i.e., uncofoundedness of representations) $Y(0), Y(1) \perp A \mid \Phi(X)$ to hold. In contrast, we allow for any, possibly confounded, representations to be input to the second stage model, $g$, analogously to the runtime confounding setting [1]. Yet, the downstream estimation (=second stage of learning) is not necessarily biased: We can still identify both representation-level CAPOs and CATE by employing the original covariate-conditional propensity score ($​​\pi^x_a$) and expected outcomes ($\mu_a^x$). We formalize this in a **new Remark 1** on the identifiability of representation-level causal quantities (see Sec. 3.2 and new Appendix C).
>
>    **Action:** We added a **new Remark 1** on the identifiability of representation-level causal quantities to the revised version of our paper (see Sec. 3.2 and new Appendix C of the **updated PDF**). It shows that CAPOs/CATE based on any subset of the covariates (e.g., based on confounded representations) are identifiable.
>
> 2. Thanks for noting this. In Eq. (10), $g$ is implicitly present, as it coincides with one of the nuisance functions ($\hat{\mu}_a^x$).
>
>    **Action:** We have rewritten Eq. 10 to make the dependency on $g$ explicit.

---

> > ### Author Response · Authors · 2024-11-22
> >
> > [2 / 2]
> >
> > 3. In our paper, we mainly explore a setting where $V$ is set to $\Phi(X)$ and _why this choice might be beneficial_. We also compare alternative choices: For example, in Sec. 4.1 (iv) (OR-learners for unconstrained representations), we discuss what happens if a target model is alternatively based on $V$ set to either the original covariates, $X$, or the predictions of the representation network, $\{\hat{\mu}_0^x(X), \hat{\mu}_1^x(X)\}$.
> > Different choices of $V$ are further explored in the experiments (see Setting A of Sec. 5), where we show that setting $V = \Phi(X)$ is highly effective, compared with other alternatives.
> >
> >    **Action:** We expanded our discussion to motivate and justify the choice of $V = \Phi(X)$ in the revised version of our manuscript.
> >
> > 4. This is an interesting question. The orthogonal losses and their minimization are first-order insensitive to the estimation of the nuisance functions. This brings two important properties of _how the first stage of estimation influences the final stage_: (1) quasi-oracle efficiency and (2) double robustness.
> >   - (1) _Quasi-oracle efficiency_  means that, even for slowly converging estimators of the nuisance functions, the minimizer of the target loss with the estimated nuisance functions would be nearly identical to the target loss with the ground truth nuisance functions. This is true for Neyman-orthogonal learners, as the estimation of the target model only depends on the higher-order errors of the nuisance functions.
> >   - (2) _Double robustness_ is achieved by DR-learners: The target model is estimated consistently if at least one of the nuisance parameters is estimated consistently.
> >
> >     We rigorously formalize these two properties with a **new Remark 2** (see Sec. 3.2 and new Appendix C).
> >
> >     **Action:** We added our **new Remark 2** (see Sec. 3.2 and new Appendix C of the **updated PDF**), which formalizes how the final estimation depends on the estimation of the nuisance functions.
> >
> >
> >
> > ## References:
> > - [1] Amanda Coston, Edward Kennedy, and Alexandra Chouldechova. Counterfactual predictions under runtime confounding. Advances in Neural Information Processing Systems, 2020.

---

> > > ### Comment · Reviewer_v7uW · 2024-11-26
> > > **Thanks for the reponse**
> > >
> > > Thanks for the detailed response. Some of the comments are partially addressed. I adjusted your score.

---

> > > > ### Author Response · Authors · 2024-11-27
> > > >
> > > > Thank you!
> > > >
> > > > We’d love to understand more about the aspects that are still missing or could be enhanced further to better align with your expectations. Are there specific areas you feel require more attention, clarity, or depth?
> > > >
> > > > Your insights are invaluable to us, and we are committed to making further improvements.

---

> ### Author Response · Authors · 2024-12-02
>
> Dear reviewer **v7uW**,
>
> We truly appreciate your time and invaluable insights!
>
> To sum it up, we hope that we have addressed all of your concerns in our rebuttal:
> - "...the authors do not provide any theoretical analysis. In addition, there is no foundation for some claims..." $\to$ We formalized many claims made in our paper by adding several remarks and propositions (see a new Appendix C).
> - "The author may want to present some evidence for some claims in the manuscript."  $\to$ We provided the proofs of all the key claims in Sec. 3 and 4 (see a new Appendix C).
> - "In addition, combining existing ideas may not be interesting enough. The writing can be much improved." $\to$ In our revised discussion (see "Choice of a target model" & "Implications" on page 10), we elaborated on the novelty of unifying representation learning and Neyman-orthogonal learners, emphasizing its non-triviality and interesting implications.
> - Question 1. $\to$ We demonstrated the identifiability of representation-level causal quantities in a new Remark 1.
> - Question 2. $\to$ We fixed Eq. 10 to make the dependency on $g$ explicit.
> - Question 3. $\to$ We extended Sec. 4.1 (iv) and the discussion (see "Choice of a target model" on page 10) where we better explained the difference between the generic $V$ and the specific $V = \Phi(X)$.
> - Question 4. $\to$ We added our new Remark 2 that formalizes how the accuracy in estimating nuisance functions affects the final estimation.
>
> As the discussion period is nearing its end, we wanted to check if you have any remaining concerns or questions we can address. If our revisions meet your expectations, we hope you might consider increasing your score.
>
> Best regards,
>
> The authors

---

### Author Response · Authors · 2024-11-22
**Response to all reviewers**

We are grateful for the reviews of our paper, we received constructive evaluation and helpful comments!   We addressed all of them in the comments below. We also uploaded a **revised version of our paper**, where we highlight key changes colored in $\color{blue}{\text{blue}}$).

Here are some of our main improvements:
- **Theoretical results**. We added Remarks 1-3 and Propositions 4-7 to formalize and prove the claims from the original submission. Therein, we showed the following:
  - We showed that the representation-conditional CAPOs/CATE are identifiable, given the covariate-conditional nuisance functions, and therefore, our OR-learners can adjust for the representation-induced confounding bias (RICB) (see a new  _Remark 1_).
  - In a new _Remark 2_, we formalized the favourable theoretical properties guaranteed by our OR-learners due to their Neyman-orthogonality (namely, double robustness and quasi-oracle efficiency).
  - We demonstrated, that under mild conditions on the representation network, the minimization of the MSE loss requires an increase of (Hölder) smoothness  (see _Remark 3_).
  - We added the proof, that a representation with $d_\phi \le 2$ is sufficient not to induce the RICB (see our new  _Proposition 4_).
  - In our new _Proposition 5_, we proved that, in order for the representation network to increase smoothness, it needs to have a Lipschitz constant of larger or equal than 1.  At the same time, in our new _Proposition 6_, we demonstrated that to achieve balancing, the converse needs to happen (namely the Lipshitz constant needs to be less or equal than 1).
  - Finally, in our new  _Proposition 7_, we showed that in the extreme case of a constant representation, our OR-learners yield efficient estimators of APOs/ATE.
- **Better contextualization of the related work**. We explained how our work differs from a meta-learners comparison provided by [Curth & van der Schaar (2021)](https://proceedings.mlr.press/v130/curth21a/curth21a.pdf). We also showed, how our work relates to the stream of literature on propensity, prognostic, and deconfounding scores.
- **Aditional visual materials**. We refactored the old Figure 6 (=new Figure 2) so that it provides a better overview of our OR-learners. Also, we added a new Figure 3, which provides a summary of how our OR-learners improve over existing representation learning methods.
- **Additional results of the experiments**. To substantiate the claims in Sec. 4, we added new experimental evidence ( _new Figure 7_). There, we showed how a representation subnetwork scales up / down the original covariate space depending on the amount of balancing applied.

With all the new changes, we have significantly improved the quality of our paper. After the rebuttal, we believe that the paper is a good fit for the ICLR 2025 conference.

---

### Meta-Review · Area_Chair_eggP · 2024-12-21

**Metareview:**

The authors propose orthogonal representation learning, or OR-learning, which is a class of Neyman-orthogonal estimators for causal quantities at the representation level. The problem is interesting and relevant to ICLR. After reading the reviews and discussions, my feeling is that this paper needs another round of review before it can be accepted. The initial submission was missing critical related work and an entire proof section, and all three reviewers pointed to these issues. The authors did put forth an extensive response and revision, but the revisions are extensive, and the proofs have only been sketched out. For this reason, the paper needs further revision (e.g. to fill out the proofs), and the revised paper needs careful review before it can be accepted.

**Additional Comments On Reviewer Discussion:**

There was an extensive discussion and revision from the authors, which I carefully reviewed. Two reviewers increased their score.

---

### Decision · Program_Chairs · 2025-01-22

Reject